# Graph Neural Networks Formed via Layer-wise Ensembles of Heterogeneous Base Models

**Jiuhai Chen** [*]                                                              *jchen169@umd.edu*
*University of Maryland*

**Jonas Mueller**                                                          *jonasmueller@csail.mit.edu*
*Cleanlab*

**Vassilis N. Ioannidis**                                              *vassilisnioannidis@gmail.com*
*Amazon*

**Tom Goldstein**                                                                 *tomg@umd.edu*
*University of Maryland*

**David Wipf**                                                               *davidwipf@gmail.edu*
*Amazon*

**Reviewed on OpenReview:** *https://openreview.net/forum?id=LO02YHxrxd*

## Abstract

Graph Neural Networks (GNNs) with numerical node features and graph structure as inputs have demonstrated superior performance on various semi-supervised learning tasks with graph data. However, the numerical node features utilized by GNNs are commonly extracted from raw data which is of text or tabular (numeric/categorical) type in most real-world applications. The best models for such data types in most standard supervised learning settings with IID (non-graph) data are not simple neural network layers and thus are not easily incorporated into a GNN. Here we propose a robust stacking framework that fuses graph-aware propagation with arbitrary models intended for IID data, which are ensembled and stacked in multiple layers. Our layer-wise framework leverages bagging and stacking strategies to enjoy strong generalization, in a manner which effectively mitigates label leakage and overfitting. Across a variety of graph datasets with tabular/text node features, our method achieves comparable or superior performance relative to both tabular/text and graph neural network models, as well as existing state-of-the-art hybrid strategies that combine the two.

## 1 Introduction

Graph datasets comprise nodes of various data types and modalities linked by edges that encapsulate non-IID conditional dependencies between them. While it is often assumed that graph neural networks (GNN) (Kipf & Welling, 2016; Veličković et al., 2017) are preferable for handling such data relative to models originally designed for IID instances, GNNs are nonetheless subject to various limitations. In particular, the best architecture may be data-set specific and require appropriately setting many attendant structural

---

*Work done during internship at Amazon Web Services Shanghai AI Lab

hyperparameters, e.g., note the complex assortment of GNN architectures that populate the top of the Open Graph Benchmark (OGB) leaderboard (Hu et al., 2020). Moreover, most GNNs implicitly assume that node features are numerical, and may struggle to remain competitive with more complex text, tabular, or composite alternatives.

In fact, with richer node feature sets it has even been observed that models tailored to IID data (which in our setting simply operate on individual node features as though they were independent of the others) can at times outperform GNNs if they are combined with simple graph propagation operations to account for the graph structure (Huang et al., 2020; Chen et al., 2021). Moreover, for graph data with text features, Chien et al. (2021) has demonstrated that leveraging a BERT Transformer in addition to a GNN can greatly improve performance. And beyond these considerations, real-world applications of ML typically involve more than just a single model, GNN or otherwise. Instead they usually require an ML pipeline composed of data preprocessing and training/tuning/aggregation of many models to achieve the best results.

In this paper, we investigate how to adapt ML pipelines designed for supervised learning with IID data (e.g., Transformers for text, gradient boosted decision trees or related for tabular data) to node classification/regression tasks with graph-structured statistical dependencies between node features. We focus on using $K$-fold bagging (Breiman, 1996), i.e. *cross-validation*, to avoid label leakage issues, with stack ensembling methods for maximal flexibility (Wolpert, 1992; Van der Laan et al., 2007). These techniques are particularly effective for achieving high accuracy across diverse IID datasets, and are utilized in many popular AutoML frameworks (Erickson et al., 2020; LeDell & Poirier, 2020; Feurer et al., 2015), but have largely been ignored within the context of graph data.

Within this context, our goal is to design a single architecture that integrates graph propagation or message passing steps and stacked ensembles of arbitrary base models to flexibly accommodate diverse node/instance types within a unified framework. In doing so, our contributions are as follows:

- We propose a framework of stack ensembling with graph propagation called **BestowGNN** for Bagged, Ensembled, Stacked Training Of Well-balanced GNNs (see Figure 1) that can *bestow* arbitrary (non-graph) base models intended for IID data with the capability of producing highly accurate node predictions in the graph (i.e., non-IID) setting.

- Using only a single, unified architecture, our proposed methodology can match or outperform bespoke dataset-specific models that top competitive leaderboards for popular node classification/regression tasks (e.g., on OGB and elsewhere completely different network architectures typically dominate the top positions for different datasets and data types).

- Label leakage is an unavoidable issue for many layer-wise training strategies (SAGN (Sun & Wu, 2021) and GAMLP (Zhang et al., 2021)). To address this potential shortcoming, we formalize how our bagging and stacking framework can effectively mitigate the label leakage issue within the graph setting using analytical tools from differential privacy. This is the first work establishing that bagging with graph-based predictors can be useful for ameliorating label leakage.

## 2 Related Work

### 2.1 From Scalability to Layer-wise Training

Currently, GNN training suffers from high computational cost as the number of layers grows. To improve the scalability, the graph sampling scheme GraphSAGE (Hamilton et al., 2017) proposes to uniformly sampling a fixed number of neighbours for a batch of nodes. Meanwhile, Cluster-GCN (Chiang et al., 2019) uses graph clustering algorithms to sample a block of nodes that form a dense subgraph and then runs SGD-based algorithms on these subgraphs. Quite differently, $L^2$-GCN (You et al., 2020) proposes a layer-wise training framework by disentangling feature aggregation and feature transformations to reduce time and memory complexity.

Other approaches include SAGN (Sun & Wu, 2021), which iteratively trains models in several stages by applying a graph structure-aware attention mechanism on node features and also combines a self-training

approach with label propagation to further improve performance. GAMLP (Zhang et al., 2021) instead proposes two attention mechanisms to explore the relation between features with different propagation steps. Both SAGN and GAMLP achieve strong performance on two large open graph benchmarks (ogbn-products and ogbn-papers100M), demonstrating the high scalability and efficiency of layer-wise training strategies. However, SAGN and GAMLP suffer from the risk of label leakage: label information is included in the enhanced training set, and can cause performance degradation if the model extracts and relies on these labels. SAGN empirically shows that sufficient propagation depth can effectively alleviate label leakage, thus they only use label information at one fixed propagation step. Meanwhile, GAMLP passes label information between propagation steps using residual connections. Wang et al. (2021) further randomly masks nodes during every training epoch to mitigate the label leakage issue. Even so, the efficacy of these methods in mitigating label leakage issue remains somewhat speculative. In contrast, by using utilizing differential privacy tools, we rigorously evaluate how our proposed bagging and stacking framework can explicitly reduce label leakage concerns in node classification settings.

## 2.2 Graph models with Multifaceted Node Features

Traditional GNN models are mostly studied for graphs with homogeneous sparse node features, and leading GNN models often fail to achieve competitive results for heterogeneous features with tabular or text node attributes (Ivanov & Prokhorenkova, 2021; Huang et al., 2020; Chen et al., 2021). To remedy this, Ivanov & Prokhorenkova (2021) jointly train Gradient Boosted Decision Trees (GBDT) and a GNN in an end-to-end fashion, demonstrating a significant increase in performance on graph data with tabular node features. Chen et al. (2021) removes the need for a GNN altogether, proposing a generalized framework for iterating boosting with parameter-free graph propagation steps that share node/sample information across edges connecting related samples. Still, both Ivanov & Prokhorenkova (2021) and Chen et al. (2021) only study regimes where tabular node features favor the using of boosted decision trees, and are unlikely to remain competitive in broader use cases, e.g., textural node features or numerical embeddings. In contrast, we demonstrate a flexible, unified framework producing competitive performance across multiple domains involving tabular, numerical, and/or textual node features. Correct and Smooth (C&S) (Huang et al., 2020) is a simple post-processing method that applies label propagation to incorporate graph information into the outputs of a learning algorithm that is otherwise agnostic to graph structure.

Turning to common GNN models that take numerical node features as inputs, one must establish a way to extract numerical embeddings from raw data such as text and images. For example, the embeddings of ogbn-arxiv data are computed by running the skip-gram model (Mikolov et al., 2013). Chien et al. (2021) proposes self-supervised learning to fully utilize correlations between graph nodes, and extracts the embeddings of three open graph benchmark datasets (ogbn-arxiv, ogbn-products and ogbn-papers100M). Chien et al. (2021) demonstrates the superior performance of these new embeddings for the Open Graph Benchmark datasets. Lin et al. (2021) proposes BertGCN, which combines the Bert model and transductive learning for text classification in an end-to-end fashion and achieves superior performance on a range of text classification tasks. However as with the boosting methods mentioned above, these techniques focus solely on one task, in contrast to our approach, where the in-built flexibility to incorporate different base models allows our method to adapt to a variety of input types with a unified overarching design.

## 3 Background

Consider an undirected graph $\mathcal{G} = (\mathcal{V}, \mathcal{E})$ with $n = |\mathcal{V}|$ nodes. The node feature matrix is denoted by $\boldsymbol{X} \in \mathbb{R}^{n \times d}$, and the corresponding node label matrix is $\boldsymbol{Y} \in \mathbb{R}^{n \times c}$ with $d$ and $c$ being the dimension of features and labels respectively. The unweighted adjacency matrix is $\boldsymbol{A} \in \mathbb{R}^{n \times n}$. For training purposes we only have access to the labels of a subset of nodes $\{\boldsymbol{y}_i\}_{i \in \mathcal{L}}$, with $\mathcal{L} \subset \mathcal{V}$. Given feature values of all nodes $\{\boldsymbol{x}_i\}_{i \in \mathcal{V}}$, label data $\{\boldsymbol{y}_i\}_{i \in \mathcal{L}}$, and the connectivity of the graph $\mathcal{E}$, the task is to predict the labels of the unlabeled nodes $\{\boldsymbol{y}_i\}_{i \in \mathcal{U}}$, with $\mathcal{U} = \mathcal{V} \setminus \mathcal{L}$. We denote the labeled dataset $\{\boldsymbol{x}_i, \boldsymbol{y}_i\}_{i \in \mathcal{L}}$ as $D_{\mathcal{L}}$ and the unlabeled dataset $\{\boldsymbol{x}_i\}_{i \in \mathcal{U}}$ as $D_{\mathcal{U}}$.

### 3.1 Bagging, Ensembling, and Stacking

For classification/regression with IID (non-graph) data, bagging, ensembling, and stacking represent practical tools that can be combined in various ways to produce more accurate predictions relative to other strategies across diverse tabular and text datasets (Shi et al., 2021; Blohm et al., 2020; Yoo et al., 2020; Fakoor et al., 2020; Bezrukavnikov & Linder, 2021; Feldman, 2021). For example, in each stacking layer of an ensemble-based architecture, bagging simply trains the same types of base models with out-of-fold predictions from the previous layer models (obtained via bagging) as extra predictive features. These base models might include various Gradient Boosted Decision Trees (Ke et al., 2017; Prokhorenkova et al., 2018), fully-connected neural networks (MLP), K Nearest Neighbors (Erickson et al., 2020), or pretrained Electra Transformer models (Clark et al., 2020). For instance, the AutoML package AutoGluon (Erickson et al., 2020) is an open-source code which is capable of exploiting these techniques.

### 3.2 Graph-Aware Propagation Layers as Energy Function Minimization

Recently there has been a surge of interest GNN architectures with layers defined in one-to-one correspondence with descent iterations that minimize a principled class of graph-regularized energy functions (Klicpera et al., 2018; Ma et al., 2020; Pan et al., 2021; Yang et al., 2021; Zhang et al., 2020; Zhu et al., 2021). In this way GNN models can benefit from the inductive bias afforded by energy function minimizers (or close approximations thereof) whose specific form can be controlled by trainable parameters. For our purposes later in Section 4, an attractive feature of this approach is that graph propagation can be conducted across an arbitrary number of layers/iterations without encountering undesirable oversmoothing effects (for a representative empirical demonstration of this capability, please see Yang et al. (2021)). The latter can degrade the performance deep GNN models by pushing all node embeddings to similar values (Li et al., 2018; Oono & Suzuki, 2020).

Following Zhou et al. (2004), one relevant energy function capable of inducing such graph-aware propagation is given by

$$\ell_Y(\boldsymbol{Y}) \triangleq (1 - \lambda) \left\| \boldsymbol{Y} - m\left(\boldsymbol{X}; \boldsymbol{\theta}\right) \right\|_{\mathcal{F}}^2 + \lambda \mathrm{tr}\left[ \boldsymbol{Y}^\top \boldsymbol{L} \boldsymbol{Y} \right], \tag{1}$$

where $\lambda \in (0, 1)$ is a weight that determines the trade-off between the two terms. $\boldsymbol{Y} \in \mathbb{R}^{n \times d}$ is a learnable $d$-dimensional embedding across $n$ nodes, and $m\left(\boldsymbol{X}; \boldsymbol{\theta}\right)$ denotes a base model (parameterized by $\boldsymbol{\theta}$) that computes an initial target embedding based on the node features $\boldsymbol{X}$. $\boldsymbol{L} \in \mathbb{R}^{n \times n}$ is the graph Laplacian of $\mathcal{G}$, meaning $\boldsymbol{L} = \boldsymbol{D} - \boldsymbol{A}$, where $\boldsymbol{D}$ represents the degree matrix.

Intuitively, the first term of (1) encourages $\boldsymbol{Y}$ to be close to initial target embedding, while the second term introduces the smoothness over the whole graph. On the positive side, the closed-form optimal solution of energy function (1) can be easily derived as

$$\widetilde{m}^*\left(\boldsymbol{X}; \boldsymbol{\theta}\right) \triangleq \arg\min_{\boldsymbol{Y}} \ell_Y(\boldsymbol{Y}) = \boldsymbol{P}^* m\left(\boldsymbol{X}; \boldsymbol{\theta}\right), \tag{2}$$

with $\boldsymbol{P}^* \triangleq (\boldsymbol{I} + \lambda \boldsymbol{L})^{-1}$. However, for large graphs the requisite inverse is impractical to compute, and alternatively iterative approximations are more practically-feasible. To this end, we may initialize as $\boldsymbol{Y}^{(0)} = m\left(\boldsymbol{X}; \boldsymbol{\theta}\right)$, and it follows that $\boldsymbol{Y}$ can be approximated by iterative descent in the direction of the negative gradient. Given that

$$\frac{\partial \ell_Y(\boldsymbol{Y})}{\partial \boldsymbol{Y}} = 2\lambda \boldsymbol{L} \boldsymbol{Y} + 2\boldsymbol{Y} - 2m\left(\boldsymbol{X}; \boldsymbol{\theta}\right), \tag{3}$$

the $t$-th iteration of gradient descent becomes

$$\boldsymbol{Y}^{(t)} = \boldsymbol{Y}^{(t-1)} - \alpha \left[ (\lambda \boldsymbol{L} + \boldsymbol{I}) \boldsymbol{Y}^{(t-1)} - m\left(\boldsymbol{X}; \boldsymbol{\theta}\right) \right], \tag{4}$$

where $\frac{\alpha}{2}$ serves as the effective step size. Considering that $\boldsymbol{L}$ is generally sparse, computation of (4) can leverage efficient sparse matrix multiplications, and we may also introduce modifications such as Jacobi preconditioning to speed convergence (Axelsson, 1996; Yang et al., 2021).

Furthermore, based on well-known properties of gradient descent, if $t$ is sufficiently large and $\alpha$ is small enough, then

$$\widetilde{m}^*\left(\boldsymbol{X}; \boldsymbol{\theta}\right) \approx \widetilde{m}^{(t)}\left(\boldsymbol{X}; \boldsymbol{\theta}\right) \triangleq \boldsymbol{P}^{(t)} \left[ m\left(\boldsymbol{X}; \boldsymbol{\theta}\right) \right], \tag{5}$$

where the operator $\boldsymbol{P}^{(t)}(\cdot)$ computes $t$ gradient steps via (4). The structure of these propagation steps, as well as related variants based on normalized modifications of gradient descent, are analogous to principled GNN layers, such as those used by GCN (Kipf & Welling, 2016), APPNP (Klicpera et al., 2018), and many others. And per the energy function association, these steps can be trained within a broader bilevel optimization framework without the risk of oversmoothing as described next.

# 4 Stack Ensembling for Graph Data (BestowGNN)

For node prediction tasks (either regression or classification), each (non-graph) base model is trained within our BestowGNN framework by simply treating each node and its label as a separate IID training example and fitting the model in the usual manner. Such a model may informatively encode tabular or text features from the nodes, but its predictions will be uniformed by the additional information available in the graph structure. To enhance such models with graph information we utilize graph-aware propagation.

## 4.1 Incorporating Graph-Aware Propagation

Let $\hat{\boldsymbol{Y}}_{\mathcal{L}}, \hat{\boldsymbol{Y}}_{\mathcal{U}}$ denote the predictions of labeled (i.e. training) nodes and unlabeled (i.e. validation/test) nodes, respectively. In node classification tasks, these may be predicted class probability vectors. Via iterative application of the update in (4), we can apply graph-aware propagation to predictions $\{\hat{\boldsymbol{Y}}_{\mathcal{L}}, \hat{\boldsymbol{Y}}_{\mathcal{U}}\}$ in order to ensure they reflect statistical dependencies between nodes encoded by the graph structure. We denote $\boldsymbol{F}^{(0)} \triangleq \{\hat{\boldsymbol{Y}}_{\mathcal{L}}, \hat{\boldsymbol{Y}}_{\mathcal{U}}\}$, and for each propagation step $t$ we compute the update $\boldsymbol{F}^{(t)} = \{\hat{\boldsymbol{Y}}_{\mathcal{L}}^{(t)}, \hat{\boldsymbol{Y}}_{\mathcal{U}}^{(t)}\}$ via (4). In our method, $\hat{\boldsymbol{Y}}$ may actually be predictions from multiple models concatenated together at each node, but the propagation procedure remains identical in this case.

## 4.2 Stack Ensembling

In stack ensembling, the predictions output by individually trained *base* models are concatenated together as features that are subsequently used to train a *stacker* model whose target is still to predict the original labels (Wolpert, 1992; Ting & Witten, 1997). A good stacker model learns how to nonlinearly combine the predictions of base models into an even more accurate prediction. This process can be iterated in multiple layers, a strategy that has been used to win high-profile prediction competitions with IID data (Koren, 2009).

In this work, we follow the stacking methodology of Erickson et al. (2020), but adapt it for graphs rather than IID data. We allow stacker models to access the original node features $\boldsymbol{X}$ by concatenating $\boldsymbol{X}$ with the base models' predictions when forming the features used to train each stacker model. To produce a final prediction for each node, we aggregate predictions from the topmost layer models via a simple weighted combination where weights are learned via the efficient Ensemble Selection technique of Caruana et al. (2004). Our base models before the first stacking layer are those which can effectively encode the original tabular or text features observed at the nodes (like Gradient Boosted Decision Trees for tabular features and Transformers for text features). Our stacker models are simply chosen as the same types of models as the base models.

## 4.3 Repeated K-fold Bagging to Mitigate Over-fitting

A problem that arises in the aforementioned stacking strategy is *label leakage*. If a base model is even slightly overfit to its training data such that its predictions memorize parts of the training labels, then subsequent stacker models will have low accuracy due to distribution shift in their features between training and inference time (their features will be highly correlated with the labels during training but not necessarily during inference). This issue is remedied by ensuring stacker models are only trained on features comprised of base model predictions on held-out nodes omitted from the base model's training set.

We achieve this while still being able to train stacker models using all labeled nodes by leveraging $K$-fold bagging (i.e. cross-validation) of all models (Van der Laan et al., 2007; Parmanto et al., 1996; Erickson et al., 2020). Here the training nodes are partitioned into $K$ disjoint chunks and $K$ copies of each (non-graph-aware) model $m$ are trained with a different data chunk held-out $\{\boldsymbol{X}^{-k}, \boldsymbol{Y}^{-k}\}_{k=1}^{K}$ held out from each copy. After training all $K$ copies of model $m$, we can produce out-of-fold (OOF) predictions $\hat{\boldsymbol{Y}}_{m}^{k}$ for each chunk $\boldsymbol{X}^{k}$

by feeding it into the model copy from which it was previously held-out. We repeat this $K$-fold bagging procedure over $N$ different random partitions of the training data to further reduce variance and distribution shift that arises in stack ensembling with bagging. Thus for a labeled training node, the OOF prediction from a model of type $m$ is averaged over $N$ different partitions (this node is held-out from exactly one model copy in each partition):

$$\hat{\boldsymbol{Y}}_{\mathcal{L}} = \left\{ \frac{1}{N} \sum_{n=1}^{N} \hat{\boldsymbol{Y}}_{m,n}^{k} \right\}_{k=1}^{K}. \tag{6}$$

Since unlabeled (validation/test) nodes were technically held-out from every model copy, we can feed them through any copy without harming stacking performance. For a particular type of model $m$, we simply make predictions $\hat{\boldsymbol{Y}}_{\mathcal{U}}$ for unlabeled nodes by averaging over all $N$ bagging repeats and all $K$ copies of the model within each repeat:

$$\hat{\boldsymbol{Y}}_{\mathcal{U}} = \frac{1}{KN} \sum_{k=1}^{K} \sum_{n=1}^{N} \hat{\boldsymbol{Y}}_{m,n}^{k}. \tag{7}$$

For IID data, this stack ensembling procedure with bagging can produce powerful predictors, both in theory (Van der Laan et al., 2007) and in practice (Erickson et al., 2020).

## 4.4 Stacking with Graph-Aware Propagation

To extend this methodology to graph data, our proposed training strategy is precisely detailed in Algorithm 1. The main idea is to apply graph-aware propagation on the predictions of models at each intermediate layer of the stack. Different amounts of propagation lead to different characteristics of the data being captured in the resulting prediction (few steps of propagation means predictions are only influenced by local neighbors, whereas many propagation steps allow predictions to be influenced by more distant nodes as well). Thus we can further enrich the feature set of our stacker models by concatenating together the predictions produced after different numbers of propagation steps. With this expanded feature set, our stacker models learn to aggregate not only the predictions of different models, but differently smoothed versions of these predictions as well. This allows the stacker model to adaptively decide how to best account for dependencies induced by the graph structure.

More precisely, given $\boldsymbol{F}^{(t)}$, the predictions (concatenated across all base model types) for labeled and unlabeled nodes after $t$ smoothing steps, then the feature input to each stacker model is given by the original node features $\boldsymbol{X}$ concatenated with $[\boldsymbol{F}^{(0)}, ..., \boldsymbol{F}^{(T)}]$. Here the predictions for labeled nodes are always OOF, obtained via bagging. Another fundamental difference between our approach and stack ensembling in the IID setting is *the use of unlabeled (test) nodes at each intermediate layer of the stack*. By including unlabeled nodes in the propagation, these nodes influence the features used to train subsequent stacker models at labeled nodes. This can even further reduce potential distribution shift in the stacker models' features between the labeled and unlabeled nodes, which ensures better generalization.

Graph machine learning models for non-IID data typically do not use bagging, seemingly because there has not been a rigorous study on the effect of bagging in relation to propagation models. Furthermore, bagging traditionally serves as a means of variance reduction which only brings limited performance benefits for large datasets (Breiman, 1996). In contrast, our stacking framework adopts bagging primarily as a means to mitigate the catastrophic effects of label leakage. While bagging can effectively mitigate label information from being directly encoded in stacker model features in the IID setting, it is not clear whether this property still holds with graph-structured dependence between nodes. A particular concern is the fact that the propagation of base model predictions across the graph implies label information is shared across the $k$-fold chunks used to hold-out some nodes from some models. In the next section, we theoretically study this issue and prove that bagging can still mitigate the effects of label leakage even in the non-IID graph setting. Our subsequent experiments (see Table 4) reveal that bagging produces substantial performance gains in practical applications of stack ensembling with graph propagation.

## 4.5 Consideration of Alternative GNN-based Message Passing

While we have adopted the energy-based message passing from Section 3.2 into our framework, it may initially seem plausible to replace these graph propagation layers with a more traditional GNN architecture. However, to incorporate GNNs in this way would require, at each stacking layer, a separate inner-loop training process, meaning multiple epochs of forward and backward passes through the GNN model to train all the parameters. In contrast, the descent steps of the graph-regularized energy functions (1) we adopt lead to efficient, parameter-free message passing, so no inner-loop training is needed. Computationally speaking, our approach is akin to just a single forward pass of a GNN, as opposed numerous forward and backward passes as would be required with training.

And incidentally, if we were to remove the GNN model parameters and simply incorporate the graph-propagation that remains, this would exacerbate the well-known oversmoothing problem mentioned in Section 3.2 whereby all node embeddings converge to similar values. In contrast, this does not occur with the energy-based graph propagation we adopt, where even an infinite number of propagation steps does not produce oversmoothing. To reiterate, this is possible because the energy minimizer itself is explicitly designed not to oversmooth, and extra propagation steps only move closer to this minimum. Please see supporting references to this effect (Ahn et al., 2022; Ma et al., 2020; Pan et al., 2021; Yang et al., 2021; Zhang et al., 2020; Zhu et al., 2021).

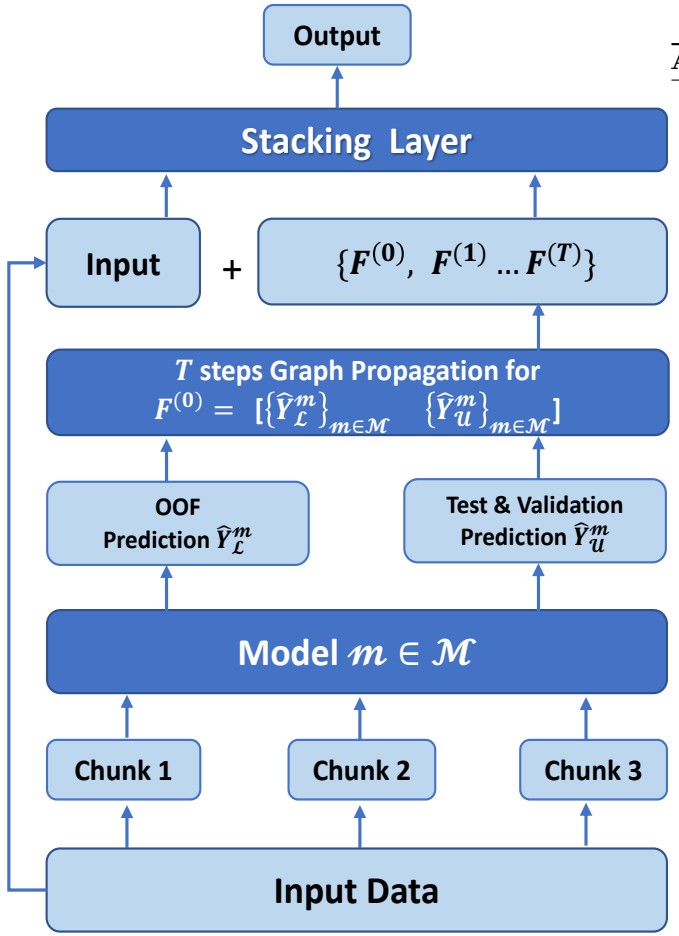

**Figure 1:** BestowGNN with a single base learner $m$, 2 stacking layers, and 3-fold bagging (repeated bagging not depicted here). The stacking layer repeats the operations depicted between it and the input data.

---

**Algorithm 1** BestowGNN Training Strategy

---

**Input:** Node features and labels $(\boldsymbol{X}, \boldsymbol{Y})$ from graph $\mathcal{G}$ with labeled (training) nodes $\mathcal{L}$ and unlabeled (validation/test) nodes $\mathcal{U}$, family of models intended for IID data $\mathcal{M}$, $L$ stacking layers, $N$-repeated, $K$-fold bagging, $T$ propagation steps.

**for** $l = 1$ **to** $L$ **do** {stacking}
    **for** $n = 1$ **to** $N$ **do** {repeated bagging}
        Randomly split data into $K$ chunks $\{\boldsymbol{X}^k, \boldsymbol{Y}^k\}_{k=1}^K$
        **for** $k = 1$ **to** $K$ **do**
            Train model $m \in \mathcal{M}$ on $\{\boldsymbol{X}^{-k}, \boldsymbol{Y}^{-k}\}$
            Make predictions $\hat{\boldsymbol{Y}}_{m,n}^k$ on OOF data $\boldsymbol{X}^k$
        **end for**
    **end for**
    **for** $m \in \mathcal{M}$ **do**
        Get OOF predictions $\hat{\boldsymbol{Y}}_{\mathcal{L}}^m$ for labeled nodes via (6)
        Get predictions $\hat{\boldsymbol{Y}}_{\mathcal{U}}^m$ for unlabeled nodes via (7)
    **end for**
    Concatenate all models' predictions:
    $\boldsymbol{F}^{(0)} \triangleq [\{\hat{\boldsymbol{Y}}_{\mathcal{L}}^m\}_{m\in\mathcal{M}}, \{\hat{\boldsymbol{Y}}_{\mathcal{U}}^m\}_{m\in\mathcal{M}}]$
    **for** $t = 0$ **to** $T$ **do** {propagation}
        Compute $\boldsymbol{F}^{(t)} = [\{\hat{\boldsymbol{Y}}_{\mathcal{L}}^m\}^{(t)}, \{\hat{\boldsymbol{Y}}_{\mathcal{U}}^m\}^{(t)}]_{m\in\mathcal{M}}$ using (4)
    **end for**
    $\boldsymbol{X} \leftarrow$ concatenate $(\boldsymbol{X}, \{\boldsymbol{F}^{(0)}, ..., \boldsymbol{F}^{(T)}\})$
**end for**
**Output:** weighted prediction $\sum_{m\in\mathcal{M}} \alpha_m \hat{\boldsymbol{Y}}_{\mathcal{U}}^m$
with $\{\alpha_m\}$ fitted via Ensemble Selection

---

# 5 Theoretical Analysis of Label Leakage

Label utilization is a common technique in which the outputs of a model are concatenated with input features and then used to train a stacking layer. Unfortunately, layer-wise training with label utilization is susceptible to the label leakage problem. Recall that previous layer-wise training methods SAGN (Sun & Wu, 2021) and GAMLP (Zhang et al., 2021) both suffer from the risk of label leakage: in the first layer, SAGN and GAMLP train the model and predict the label of training points, then predicted labels are included in the enhanced training set. In the second layer, they use the enhanced training set as the new feature to train a new model. Notice here including predicted labels as training features can cause performance degradation if the model extracts and relies on these predicted labels. The second layer training upon first-layer predictions could amplify over-fitting issues and introduce the covariate shift at test time. Although prior work (Sun & Wu, 2021; Zhang et al., 2021) has mentioned heuristic ways to address label leakage via graph propagation, it is unclear how generally applicable this strategy is in practice. Moreover, there is a natural trade-off between avoiding label leakage via graph propagation, and well-known oversmoothing effects in GNN models.

In this section we employ a powerful theoretical tool, Differential Privacy (Mironov, 2017), to showcase the advantage of bagging in our proposed BestowGNN. Our analysis will show that BestowGNN enjoys strong generalization under the Rényi Differential Privacy framework. In fact this is the first work that establishes that bagging in graph predictors is useful and mitigates label leakage. Specifically, BestowGNN can preserve the privacy (or information sharing) of labels between bags, that would otherwise be compromised by graph propagation.

To this end, we first introduce the definition of Rényi Differential Privacy, which is a relaxation of Differential Privacy based on the Rényi Divergence.

**Definition 1.** *(Rényi Differential Privacy (Mironov, 2017)). Consider a randomized algorithm $\mathcal{M}$ mapping from $\mathcal{D}$ to a real-value $\mathcal{R}$. Such an algorithm is said to have $\epsilon$-Rényi Differential Privacy of order $\alpha$ if for any $D, D' \in \mathcal{D}$ with $d_H(D, D') = 1$, where $d_H$ is the Hamming distance ($D, D'$ are also referred to as adjacent datasets), we have that*

$$D_\alpha(\mathcal{M}(D)||\mathcal{M}(D'))$$
$$\triangleq \frac{1}{\alpha - 1} \log E_{x \sim \mathcal{M}(D')} \left( \frac{\mathcal{M}(D)}{\mathcal{M}(D')} \right)^\alpha \leq \epsilon. \tag{8}$$

In plain words, this definition establishes that the output of an algorithm does not change significantly, as measured by the Rényi divergence $D_\alpha(\mathcal{M}(D)||\mathcal{M}(D'))$, when the data changes slightly. The idea behind this framework is that if each individual data sample has only a small effect on the resulting model, the model cannot be used to infer information about any single individual.

We then have the following result:

**Theorem 1.** *Assume base model $m$ is a multi-layer (two-layer) perceptron and that node features $\boldsymbol{X}$ are sampled from a multivariate Gaussian as in (Jia & Benson, 2021):*

$$\boldsymbol{X} \sim \mathcal{N}(\boldsymbol{0}, \boldsymbol{\Gamma}^{-1}), \qquad \boldsymbol{\Gamma} = c_1 \boldsymbol{I}_n + c_2 \boldsymbol{L},$$

*where $\boldsymbol{I}_n$ is an identity matrix and $\boldsymbol{L}$ is the normalized graph Laplacian. Here $c_1$ controls a noise level and $c_2$ the smoothness over the whole graph. $\boldsymbol{E}(x_0; D_{\mathcal{L}})$ and $\boldsymbol{F}(x_0; D_{\mathcal{L}})$ are predictions produced by BestowGNN for a data point $x_0$ with and without bagging mode, respectively. If $\boldsymbol{E}$ has sensitivity 1 and lower magnitude bound $L$, i.e., for any two adjacent $D, D' \in D : |\boldsymbol{E}(\boldsymbol{x}_0; D) - \boldsymbol{E}(\boldsymbol{x}_0; D')| \leq 1$ and $|\boldsymbol{E}| \geq L$, then $\boldsymbol{E}$ satisfies $(\frac{1}{2}, \frac{1}{4\sigma^2 L^2} + \frac{1}{2L^2})$-Rényi Differential Privacy, where $\sigma^2$ depends on graph structure $\mathcal{G}$. Meanwhile, $\boldsymbol{F}$ has no privacy guarantee, i.e., the Rényi differential privacy loss (8) is unbounded.*

The proof is deferred to the appendix. Theorem 1 indicates that bagging with graph propagation can well preserve the privacy of $D_{\mathcal{L}} = \{\boldsymbol{x}_i, \boldsymbol{y}_i\}_{i \in \mathcal{L}}$ between different chunks while non-bagging would have a high risk of leaking the information of $D_{\mathcal{L}}$. For layer-wise training with label utilization, the output of the model $\boldsymbol{E}(x_0; D_{\mathcal{L}})$ is concatenated with input features and then used to train next stacking layer, and bagging can effectively mitigate the label leakage issue since the information of true label is well preserved at the first layer, while no-bagging exposes the true labels and can lead to over-fitting issue for next stacking layer.

## 6    Experiments

**Datasets.**    We study the effectiveness of our approach by comparing performance against a variety of baselines in node regression and classification tasks. For node regression with **tabular node features**, we consider four real-world graph datasets used for benchmarking by Ivanov & Prokhorenkova (2021): House, County, VK and Avazu. As node classification tasks, we adopt one dataset with **numerical features**: Reddit; and two datasets with **raw text features**: OGB-Arxiv and OGB-Products. We also consider the OGB datasets using original OGB provided features. More details about the datasets are provided in the appendix.

**Baseline approaches.**    We compare our method against various baselines, starting with purely tabular baseline models or language models where the graph structure is ignored. Our first baseline is **Autogluon** (Erickson et al., 2020), an AutoML system for IID tabular or text data that is completely unaware of the graph structure (here we simply treat nodes as IID). Next, we consider **AutoGluon + C&S**, which performs Correct and Smooth (Huang et al., 2020) as a post hoc processing step on top of AutoGluon's predictions, in order to at least account for the graph structure during inference. For node regression tasks we also consider some popular GNN models: **GCN** (Kipf & Welling, 2016), **GAT** (Veličković et al., 2017), and a hybrid strategy **BGNN** (Ivanov & Prokhorenkova, 2021), which combines Gradient Boosted Decision Trees (also a model intended for IID data) with GNNs via end-to-end training in a manner that is graph-aware.

For node classification using Reddit with original numerical features, we also compare with **GraphSAGE** (Hamilton et al., 2017) and **PCAPass + Tree** (Sadowski et al., 2022), which combines PCA and message passing to generate node embeddings and leverages tree-based model for node classification.

For OGB-Arxiv and OGB-Products with original OGB features (pre-computed low-dimensional text embeddings as node features provided by OGB), we consider standard GNN models: **GCN** (Kipf & Welling, 2016) and **GAT** (Veličković et al., 2017) variants, and **Ensemble GCN**, a natural baseline/competitor which divides all training nodes into $K$ chunks, trains a GCN model for each chunk and then ensembles the results. We also compare against SOTA models for OGB-Arxiv and OGB-Products from the OGB leaderboard (restricted to original features, no text augmentation) when we initially started the experiments (AGDN+BoT+self-KD+C&S for Arxiv and GAMLP+RLU+SCR+C&S for Products).

For OGB-Arxiv and OGB-Products with raw text as node features (these are necessarily more informative than the compressed original OGB features), we chose the method that topped the respective OGB leaderboard at the time of our submission: this was TAPE+RevGAT (He et al., 2023) for OGB-Arxiv and GLEM+GIANT+SAGN+SCR (Zhao et al., 2023) for OGB-Products. To assess the adaptability of these two approaches, we retrain both models with standard hyperparameter tuning on the *opposite* dataset for which they topped the leaderboad, meaning we also tested TAPE+RevGAT on OGB-Products and GLEM+GIANT+SAGN+SCR on OGB-Arxiv. This is a reasonable scenario since both models are designed to handle text features, and yet given their complex, composite natures, it is unclear how well they might reliably transfer to different datasets.

**BestowGNN details.**    We evaluate our method **BestowGNN**, which incorporates base models and graph information through propagation operations within each stacking layer. For the base models, we consider LightGBM boosted Tress (GBM) (Ke et al., 2017), CatBoost boosted trees (CAT) (Prokhorenkova et al., 2018), fully-connected neural networks (NN), Extremely Randomized Trees (RT), Random Forests (RF), K Nearest Neighbors (KNN), Label Propagation (LP) (Huang et al., 2020) and Transformer with electra pretrained model (Text) (Training epoch is 12) (Clark et al., 2020). For the first layer, we keep the typical models, for example, Gradient Boosted Decision Trees for Tabular data, Transformer models for text data. For second stacking layer, we use all of models except extremely low-efficient models for large dataset, for example, KNN and Catboost slow down the training procedure for the OGB-products dataset. All details regarding the base models, as well as the minor/standard hyperparameter tuning used by BestowGNN to fit all datasets, can be found in the appendix.

**Results.**    In Table 1 we present the results for the node regression task with tabular node features. The baseline GNN models are challenged by the tabular node features. AutoGluon is an ensemble of various base models (e.g., Gradient Boosted Decision Trees, fully-connected neural networks) intended for IID data without

| Data set | House | County | Vk | Avazu |
|---|---|---|---|---|
| GCN | 0.63 ± 0.01 | 1.48 ± 0.08 | 7.25 ± 0.19 | 0.1141 ± 0.02 |
| GAT | 0.54 ± 0.01 | 1.45 ± 0.06 | 7.22 ± 0.19 | 0.1134 ± 0.01 |
| BGNN | 0.50 ± 0.01 | 1.26 ± 0.08 | 6.95 ± 0.21 | 0.109 ± 0.01 |
| AutoGluon | 0.618 ± 0.01 | 1.379 ± 0.08 | 7.176 ± 0.21 | 0.117 ± 0.018 |
| AutoGluon + C&S | 0.477 ± 0.01 | 1.162 ± 0.09 | 6.995 ± 0.21 | 0.107 ± 0.015 |
| BestowGNN | **0.467 ± 0.007** | **1.145 ± 0.083** | **6.918 ± 0.220** | **0.105 ± 0.013** |

**Table 1:** Mean squared error of different methods for four different node regression datasets (The top model is boldfaced, the second-best model is underlined).

**Table 2:** Node classification accuracy for Reddit with **numerical** node features (The best model is boldfaced, the second-best model is underlined).

| Method | Reddit |
|---|---|
| PCAPass + XGBoost | 96.26 ± 0.02 |
| GraphSAGE | 95.40 ± 0.22 |
| AutoGluon | 95.83 ± 0.00 |
| AutoGluon + C&S | 96.00 ± 0.00 |
| BestowGNN | **96.44 ± 0.00** |

considering graph structure. We observe that Autogluon + C&S outperforms Autogluon, demonstrating that graph information can greatly boost the performance of models intended for IID data. Incorporating the graph structure at each stacking layer, our BestowGNN method performs better than prior baselines on all datasets. While the performance difference relative to the top performing alternatives may be modest at times, as we will soon observe, BestowGNN is consistently at or near the top spanning all datasets and scenarios. As such, a cumulative case can be made for the efficacy of BestowGNN relative to other approaches that excel only in limited domains.

Next, Table 2 shows that BestowGNN outperforms the baselines for the Reddit dataset with numerical embeddings, while PCAPass + XGBoost is second. Turning to Table 3, we display results for OGB-Arxiv and OGB-Products, and highlight the superior performance of models trained using raw text features as would be expected. Of particular note though is the performance of the top OGB leaderboard methods when retrained on a different dataset. Specifically, while GLEM+GIANT+SAGN+SCR may be best on OGB-Products (at the time of our submission), when transferred to OGB-Arxiv there is a considerable drop-off (its accuracy of 77.50 here would only rank 16th on the current Arxiv leaderboard). Similarly, although TAP+RevGAT may have been tops for OGB-Arxiv, it is not especially competitive on OGB-Products (its accuracy of 82.16 places only 30th as of the current OGB-Products leaderboard). We also note that AutoGluon+C&S, which was competitive on the tabular benchmarks in Table 1, is far less competitive using OGB data, especially on OGB-Products. In contrast, BestowGNN uses essentially the same core architecture, which includes the incorporation of graph information within each stacking layer, to fit all datasets spanning Tables 1, 2, and 3.

**Ablation.** The key ingredients of our framework are bagging/ensembling and graph propagation. Table 4 shows an ablation study involving these components using OGB-Arxiv with original OGB embeddings. From the above results, we observe that in each case the training performance under the no-bagging setting is always higher than bagging as would be expected if some degree of overfitting were occurring. In contrast, on the test (and validation) sets, the situation is reversed and no-bagging now outperforms bagging. This indicates that bagging has helped to mitigate some of the effects of overfitting by reducing the gap between training and testing accuracy. For additional ablations, please see the appendix.

**Table 3:** Node classification accuracy for OGB-Arxiv and OGB-Products (The best model is boldfaced, the second-best model is underlined). Rows labeled TEXT contain involve models trained on raw text embeddings/features, while those labeled OGB indicate models trained on precomputed numerical embeddings provided by OGB as node features. As of the time of our submission TAPE+RevGAT was the top performing model on OGB-Arxiv while GLEM+GIANT+SAGN+SCR was the top performing model on OGB-Products; however, in both cases relative performance degrades when shifting to a different dataset, despite the similar text-based modeling scenarios. Meanwhile, BestowGNN maintains competitive performance spanning these and other datasets/scenarios.

OGB-Arxiv

| Feature | Method | Test Acc (Validation) |
|---|---|---|
| OGB | GCN | 73.06 ± 0.24 (74.42 ± 0.12) |
| | GAT + C&S | 73.86 ± 0.14 (74.84 ± 0.07) |
| | AGDN+BoT+self-KD+C&S | 74.31 ± 0.14 (75.18 ± 0.09) |
| | Ensemble GCN | 73.22 ± 0.12 (74.64 ± 0.01) |
| TEXT | GLEM+GIANT+SAGN+SCR | 75.50 ± 0.11 (76.87 ± 0.09) |
| | TAPE+RevGAT | **77.50 ± 0.12 (77.85 ± 0.16)** |
| TEXT | AutoGluon | 73.05 ± 0.00 (74.33 ± 0.00) |
| | AutoGluon + C&S | 75.34 ± 0.00 (76.67 ± 0.00) |
| TEXT | BestowGNN | 76.19 ± 0.02 (77.25 ± 0.05) |

OGB-Products

| Feature | Method | Test Acc (Validation) |
|---|---|---|
| OGB | DeeperGCN + FLAG | 81.93 ± 0.31 (91.03 ± 0.01) |
| | GAT + FLAG | 81.76 ± 0.45 (92.51 ± 0.06) |
| | GAMLP+RLU+SCR+C&S | 85.20 ± 0.08 (93.04 ± 0.05) |
| | Ensemble GAT | 80.01 ±0.20 (93.24 ± 0.05) |
| TEXT | GLEM+GIANT+SAGN+SCR | **87.37 ± 0.06 (94.00 ± 0.03)** |
| | TAPE+RevGAT | 82.16 ± 0.28 (92.10 ± 0.09) |
| TEXT | AutoGluon | 77.10 ± 0.06 (91.78 ± 0.03) |
| | AutoGluon + C&S | 79.03 ± 0.12 (93.62 ± 0.03) |
| TEXT | BestowGNN | 85.48 ± 0.03 (93.93 ± 0.02) |

**Table 4:** BestowGNN ablation study with (✓) and without bagging (✗). Here $T$ is the number of graph propagation steps, thus $T = 0$ represents a baseline model that completely ignores graph structure.

| STEP $T$ | TRAIN | | VALIDATION | | TEST | |
|---|---|---|---|---|---|---|
| | ✓ | ✗ | ✓ | ✗ | ✓ | ✗ |
| 0 | 0.64 | 0.68 | 0.58 | 0.56 | 0.56 | 0.54 |
| 1 | 0.76 | 0.78 | 0.67 | 0.66 | 0.67 | 0.65 |
| 2 | 0.77 | 0.78 | 0.70 | 0.68 | 0.70 | 0.68 |
| 3 | 0.77 | 0.78 | 0.71 | 0.69 | 0.70 | 0.68 |
| 4 | 0.77 | 0.79 | 0.71 | 0.70 | 0.70 | 0.69 |
| 50 | 0.79 | 0.81 | 0.72 | 0.71 | 0.71 | 0.69 |

**Table 5:** Training time tested on AWS g4dn.12xlarge machine.

| DATASET | BASE MODEL | TIME(S) |
|---|---|---|
| HOUSE | GBM, NN | 52 |
| COUNTY | GBM, NN | 18 |
| VK | GBM, NN | 119 |
| AVAZU | GBM, NN | 15 |
| OGB-ARXIV | NN | 199 |
| OGB-PRODUCTS | NN | 837 |

**Computing cost.** The computing cost depends on the ensemble models we select (e.g., transformer models can take more computing resources relying on the implementation, including more emsemble models leads to more computing cost). As a result, it is difficult to consistently measure the training/inference time or memory consumption. Nonetheless, the computing cost lies in a competitive range since the integration of the bagging and ensembling parts key to our model can be efficiently implemented, e.g., via open source packages like AutoGluon that we used. In Table 6, we present the training time for different datasets with basic ensemble models on an AWS g4dn.12x Large machine.

## 7    Discussion

While real-world graph data come with heterogeneous feature types, existing GNN models are primarily suited for (adequately preprocessed) numerical features. For IID supervised learning, it is well-known that the best models for different feature types vary based on dataset and data-type, and that a learning system aiming to output good predictions across a variety of datasets should leverage a heterogeneous collection of different types of models (Erickson et al., 2020). There is little reason the situation should be different for graph data. In this paper, we demonstrate the first working system that can utilize arbitrary heterogeneous collections of models for arbitrary graph datasets with heterogeneous feature-types (numerical, categorical, text). This is achieved by means of a novel graph-aware stack ensembling technique that takes the graph structure into account without restricting how individual models are trained. Our graph-aware propagation techniques leverage specific properties of stack ensembling that allow our proposed methodology to outperform both many complex GNNs as well as existing approaches in which propagation is only applied to the predictions output by an IID base model (e.g., AutoGluon+C&S, etc.).

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

# Appendix:

## A  Proof of Theorem 1.

**Preliminary 1.**  Firstly, we derive the format of $\boldsymbol{E}(\boldsymbol{x}_0; D_{\mathcal{L}})$ and $\boldsymbol{F}(\boldsymbol{x}_0; D_{\mathcal{L}})$. Suppose BestowGNN randomly splits the labeled nodes $D_{\mathcal{L}}$ into 2 disjoint chunks $D_1 = \{\boldsymbol{X}_1, \boldsymbol{Y}_1\}, D_2 = \{\boldsymbol{X}_2, \boldsymbol{Y}_2\}$. BestowGNN trains a model $m \in \mathcal{M}$ with a different data chunk held-out. Model $m$ is defined by a set of parameters collected in $\boldsymbol{\theta}$ namely, which is defined as $m(\boldsymbol{X}; \boldsymbol{\theta})$. In the following, we will express the predicted labels from model $m$ under the bagging and non-bagging settings. We compare the predicted labels under both settings and establish that our bagging solution is less amenable to label leakage.

The model $m$ will learn different parameters for each chunk and those are denoted as $\boldsymbol{\theta}_1$ for the chunk I and $\boldsymbol{\theta}_2$ for the chunk II, namely $\boldsymbol{\theta}_1 = \boldsymbol{\theta}(D_1)$ and $\boldsymbol{\theta}_2 = \boldsymbol{\theta}(D_2)$. Next, BestowGNN produces prediction $\hat{\boldsymbol{Y}}_1, \hat{\boldsymbol{Y}}_2$ on out-of-fold data, i.e., $\hat{\boldsymbol{Y}}_1 = m(\boldsymbol{X}_1; \boldsymbol{\theta}_2)$ and $\hat{\boldsymbol{Y}}_2 = m(\boldsymbol{X}_2; \boldsymbol{\theta}_1)$. The prediction for unlabeled nodes is $\hat{\boldsymbol{Y}}_{\mathcal{U}} = \frac{1}{2}[m(\boldsymbol{X}_{\mathcal{U}}; \boldsymbol{\theta}_1) + m(\boldsymbol{X}_{\mathcal{U}}; \boldsymbol{\theta}_2)]$ as explained in (7). Consider one data point $\boldsymbol{x}_0$ from the unlabeled dataset $D_{\mathcal{U}}$, the prediction of $\boldsymbol{x}_0$ is given by $\hat{\boldsymbol{y}}_0 = \frac{1}{2}[m(\boldsymbol{x}_0; \boldsymbol{\theta}_1) + m(\boldsymbol{x}_0; \boldsymbol{\theta}_2)]$. Next, we perform one step graph-aware propagation on $\hat{\boldsymbol{y}}_0$.

$$
\begin{aligned}
\hat{\boldsymbol{y}}_0^{(1)} &= \sum_{u \in \mathcal{N}(\boldsymbol{x}_0) \cap D_{\mathcal{U}}} \hat{\boldsymbol{y}}_u + \sum_{v \in \mathcal{N}(\boldsymbol{x}_0) \cap D_1} \hat{\boldsymbol{y}}_v + \sum_{w \in \mathcal{N}(\boldsymbol{x}_0) \cap D_2} \hat{\boldsymbol{y}}_w \\
&= \sum_{u \in \mathcal{N}(\boldsymbol{x}_0) \cap D_{\mathcal{U}}} \frac{1}{2}[m(\boldsymbol{x}_u; \boldsymbol{\theta}_1) + m(\boldsymbol{x}_u; \boldsymbol{\theta}_2)] + \sum_{v \in \mathcal{N}(\boldsymbol{x}_0) \cap D_1} m(\boldsymbol{x}_v; \boldsymbol{\theta}_2) + \sum_{w \in \mathcal{N}(\boldsymbol{x}_0) \cap D_2} m(\boldsymbol{x}_w; \boldsymbol{\theta}_1),
\end{aligned}
\tag{9}
$$

where $\hat{\boldsymbol{y}}_0^{(1)}$ is the aggregated results from one-hop neighbor $\mathcal{N}(\boldsymbol{x}_0)$, which may belongs to $D_{\mathcal{U}}, D_1$ and $D_2$.

Next, we consider the no-bagging mode, where the predictions of $\boldsymbol{X}_1, \boldsymbol{X}_2$ are changed into $\widetilde{\boldsymbol{Y}}_1 = m(\boldsymbol{X}_1; \boldsymbol{\theta}_1)$ and $\widetilde{\boldsymbol{Y}}_2 = m(\boldsymbol{X}_2; \boldsymbol{\theta}_2)$. Notice that with bagging mode we use the parameters from a different bag, while without bagging we use the parameters from the same bag. The prediction of the test point $\boldsymbol{x}_0$ is once again $\widetilde{\boldsymbol{y}}_0 = \frac{1}{2}[m(\boldsymbol{x}_0; \boldsymbol{\theta}_1) + m(\boldsymbol{x}_0; \boldsymbol{\theta}_2)]$, which is identical to the bagging mode. We perform the same graph-aware propagation on $\widetilde{\boldsymbol{y}}_0$.

$$
\begin{aligned}
\widetilde{\boldsymbol{y}}_0^{(1)} &= \sum_{u \in \mathcal{N}(\boldsymbol{x}_0) \cap D_{\mathcal{U}}} \widetilde{\boldsymbol{y}}_u + \sum_{v \in \mathcal{N}(\boldsymbol{x}_0) \cap D_1} \widetilde{\boldsymbol{y}}_v + \sum_{w \in \mathcal{N}(\boldsymbol{x}_0) \cap D_2} \widetilde{\boldsymbol{y}}_w \\
&= \sum_{u \in \mathcal{N}(\boldsymbol{x}_0) \cap D_{\mathcal{U}}} \frac{1}{2}[m(\boldsymbol{x}_u; \boldsymbol{\theta}_1) + m(\boldsymbol{x}_u; \boldsymbol{\theta}_2)] + \sum_{v \in \mathcal{N}(\boldsymbol{x}_0) \cap D_1} m(\boldsymbol{x}_v; \boldsymbol{\theta}_1) + \sum_{w \in \mathcal{N}(\boldsymbol{x}_0) \cap D_2} m(\boldsymbol{x}_w; \boldsymbol{\theta}_2).
\end{aligned}
\tag{10}
$$

Next, we compare the terms among the predicted labels from the two settings, namely (9) and (10). The first term $\sum_{u \in \mathcal{N}(\boldsymbol{x}_0) \cap D_{\mathcal{U}}} \frac{1}{2}[m(\boldsymbol{x}_u; \boldsymbol{\theta}_1) + m(\boldsymbol{x}_u; \boldsymbol{\theta}_2)]$ is the same for (9) and (10) and can be cancelled. In order to facilitate the exposition of the theoretical contributions we will define functions for the different terms in (9) and (10). We define $\boldsymbol{E}(\boldsymbol{x}_0; D_{\mathcal{L}})$, that is a function formulating the relation between training data $D_{\mathcal{L}}$ and the prediction for test data $\boldsymbol{x}_0$ under bagging mode.

$$
\boldsymbol{E}(\boldsymbol{x}_0; D_{\mathcal{L}}) := \sum_{v \in \mathcal{N}(\boldsymbol{x}_0) \cap D_1} m(\boldsymbol{x}_v; \boldsymbol{\theta}(D_2)) + \sum_{w \in \mathcal{N}(\boldsymbol{x}_0) \cap D_2} m(\boldsymbol{x}_w; \boldsymbol{\theta}(D_1)).
\tag{11}
$$

Similarly, we define the function $\boldsymbol{F}(\boldsymbol{x}_0; D_{\mathcal{L}})$ formulating the relation between training data $D_{\mathcal{L}}$ and the prediction for test data $\boldsymbol{x}_0$ under the no-bagging mode:

$$
\boldsymbol{F}(\boldsymbol{x}_0; D_{\mathcal{L}}) := \sum_{v \in \mathcal{N}(\boldsymbol{x}_0) \cap D_1} m(\boldsymbol{x}_v; \boldsymbol{\theta}(D_1)) + \sum_{w \in \mathcal{N}(\boldsymbol{x}_0) \cap D_2} m(\boldsymbol{x}_w; \boldsymbol{\theta}(D_2)).
\tag{12}
$$

Notice here $\boldsymbol{\theta}(D_1)$ is the model parameters of Chunk I involving information of true label $\boldsymbol{Y}_1$. We aim to examine bagging and stacking strategies effectively preserve the information of label $\boldsymbol{Y}_1$ via introducing

randomness to the function $\boldsymbol{E}(\boldsymbol{x}_0; D_{\mathcal{L}})$ while $\boldsymbol{F}(\boldsymbol{x}_0; D_{\mathcal{L}})$ has high risk of leaking the information of true label $\boldsymbol{Y}_1$.

We first reiterate the definition of Rényi Differential Privacy.

**Definition 1.** *(Rényi Differential Privacy (Mironov, 2017)). Consider a randomized algorithm $\mathcal{M}$ mapping from $\mathcal{D}$ to real-value $\mathcal{R}$. Such an algorithm is said to have $\epsilon$-Rényi Differential Privacy of order $\alpha$ ($\alpha > 1$) if any $D, D' \in \mathcal{D}$ with $d_H(D, D') = 1$, where $d_H$ is the Hamming distance ($D, D'$ are also referred to as adjacent datasets):*

$$D_\alpha(\mathcal{M}(D)||\mathcal{M}(D')) = \frac{1}{\alpha - 1} \log E_{x \sim \mathcal{M}(D')} \left( \frac{\mathcal{M}(D)}{\mathcal{M}(D')} \right)^\alpha \le \epsilon. \tag{13}$$

To proceed in a quantifiable way, we rely on some preliminary results for Rényi Differential privacy and generative model for graph learning algorithms.

**Proposition 1.** *Rényi differential privacy is preserved by post-processing (Mironov, 2017). If $F(\cdot)$ has $\epsilon$-Rényi Differential Privacy, then for any randomized or deterministic function $g$, $g(F(\cdot))$ satisfies $\epsilon$-Rényi Differential Privacy.*

**Proposition 2.** *The closed-form expression of the Rényi divergence between any two Gaussian distributions is given by $D_\alpha(\mathcal{N}(\mu_0, \sigma_0^2)||\mathcal{N}(\mu_1, \sigma_1^2)) = \frac{\alpha(\mu_1 - \mu_0)^2}{2\sigma_\alpha^2} + \frac{1}{1-\alpha} \ln \frac{\sigma_\alpha}{\sigma_0^{1-\alpha}\sigma_1^\alpha}$, provided that $\sigma_\alpha^2 = (1 - \alpha)\sigma_0^2 + \alpha\sigma_1^2 > 0$ (Van Erven & Harremos, 2014).*

**Proposition 3.** *Assume $f$ has sensitivity 1 and lower magnitude bound $L$, i.e., for any pair of adjacent datasets $D, D' \in \mathcal{D}$: $|f(D) - f(D')| \le 1$ and $|f| \ge L$, and define the Gaussian multiplicative mechanism*

$$\boldsymbol{GM}_{\mu,\sigma} f(D) = f(D)\mathcal{N}(\mu, \sigma^2).$$

*Then $\boldsymbol{GM}_{\mu,\sigma} f$ satisfies $(\frac{1}{2}, \frac{1}{4\sigma^2 L^2} + \frac{1}{2L^2})$-Rényi Differential Privacy.*

*Proof.* According to Proposition (2):

$$D_{1/2}\left(\mathcal{N}(f(D) + \mu, f^2(D)\sigma^2)||\mathcal{N}(f(D') + \mu, f^2(D')\sigma^2)\right)$$

$$= \frac{(f(D) - f(D'))^2}{2\sigma^2(f^2(D) + f^2(D'))} + \ln[\frac{1}{2}(f^2(D) + f^2(D'))] - \ln|f(D)| - \ln|f(D')|$$

$$= \frac{1}{2\sigma^2} - \frac{f(D)f(D')}{\sigma^2(f^2(D) + f^2(D'))} + \ln[\frac{1}{2}(f^2(D) + f^2(D'))] - \ln|f(D)f(D')|$$

$$= \frac{1}{2\sigma^2} - \frac{f(D)f(D')}{\sigma^2(f^2(D) + f^2(D'))} + \ln|\frac{f^2(D) + f^2(D')}{2f(D)f(D')}|$$

$$\le \frac{1}{2\sigma^2}\frac{1}{f^2(D) + f^2(D')} + \ln(\frac{1}{2|f(D)f(D')|} + 1)$$

$$\le \frac{1}{4\sigma^2 L^2} + \ln(\frac{1}{2L^2} + 1)$$

$$\le \frac{1}{4\sigma^2 L^2} + \frac{1}{2L^2}.$$

The first inequality follows from $|f(D) - f(D')| \le 1$, take square for both side $f^2(D) + f^2(D') \le 1 + 2f(D)f(D')$. Then we have $\frac{1}{2\sigma^2} - \frac{f(D)f(D')}{\sigma^2(f^2(D) + f^2(D'))} \le \frac{1}{2\sigma^2}\frac{1}{f^2(D) + f^2(D')}$ and $\frac{f^2(D) + f^2(D')}{2|f(D)f(D')|} \le \frac{1}{2|f(D)f(D')|} + 1$, the first inequality holds. $\square$

**Proposition 4.** *If $f$ has sensitivity 1, i.e., for any pair of adjacent datasets $D, D' \in \mathcal{D}$: $|f(D) - f(D')| \le 1$. Define the Gaussian additive mechanism*

$$\boldsymbol{GA}_\sigma f(D) = f(D) + \mathcal{N}(0, \sigma^2),$$

*then Gaussian additive mechanism $\boldsymbol{GA}_\sigma f$ satisfies $(\alpha, \frac{\alpha}{2\sigma^2})$-Rényi Differential Privacy (Mironov, 2017).*

**Proposition 5.** *Consider a multivariate Gaussian distribution, and the random variables are partitioned into two groups $(z_P, z_Q)$, the distribution is block matrix format*

$$\begin{pmatrix} z_P \\ z_Q \end{pmatrix} \sim \mathcal{N} \left( \begin{bmatrix} \bar{z}_P \\ \bar{z}_Q \end{bmatrix}, \begin{bmatrix} \Gamma_{PP} & \Gamma_{PQ} \\ \Gamma_{QP} & \Gamma_{QQ} \end{bmatrix}^{-1} \right),$$

*where $\begin{bmatrix} \Gamma_{PP} & \Gamma_{PQ} \\ \Gamma_{QP} & \Gamma_{QQ} \end{bmatrix}$ is precision (inverse covariance) matrix. Then the marginal and conditional distribution can be written as*

$$z_P \sim \mathcal{N} \left( \bar{z}_P, (\Gamma_{PP} - \Gamma_{PQ}\Gamma_{QQ}^{-1}\Gamma_{QP})^{-1} \right), \tag{14}$$

$$z_P | z_Q = z_Q \sim \mathcal{N} \left( \bar{z}_P - \Gamma^{-1}\Gamma_{PQ}(z_Q - \bar{z}_Q) \right). \tag{15}$$

Before proceeding to our specific results in the main paper, we also need to describe the graph setting.

**Preliminary 2.** Let $\mathcal{G} = (V, E)$ be an undirected graph, where $V$ is the set of $n$ nodes and $E$ is the set of edges. The adjacency matrix of $\mathcal{G}$ is $W \in \mathcal{R}^{n \times n}$, the diagonal degree matrix is $D \in \mathcal{R}^{n \times n}$. The normalize graph Laplacian can be written as $N = I - D^{-1/2}WD^{-1/2} = I - S$. We use $X \in \mathcal{R}^{n \times p}$ for the feature matrix, where p is the dimension of features. We assume all vertex features $X$ are jointly sampled from a multivariate Gaussian distribution (Jia & Benson, 2021), namely

$$X \sim \mathcal{N}(0, \Gamma^{-1}), \qquad \Gamma = c_1 I_n + c_2 N, \tag{16}$$

where $I_n$ is identical matrix, $N$ is normalized graph Laplacian. Here $c_1$ controls noise level and $c_2$ controls the smoothness over the whole graph.

We now proceed to our specific results in the main paper.

**Theorem 1.** *Assume base model m to be a multi-layer (two-layer) perceptron and node features $X$ is sampled from a multivariate Gaussian as in Jia & Benson (2021):*

$$X \sim \mathcal{N}(0, \Gamma^{-1}), \qquad \Gamma = c_1 I_n + c_2 L,$$

*where $I_n$ is an identity matrix and $L$ is the normalized graph Laplacian. Here $c_1$ controls noise level and $c_2$ controls the smoothness over the whole graph. $E(x_0; D_{\mathcal{L}})$ and $F(x_0; D_{\mathcal{L}})$ are predictions produced by BestowGNN for a data point $x_0$ with and without bagging mode, respectively. If $E$ has sensitivity 1 and lower magnitude bound L, i.e., for any two adjacent $D, D' \in D : |E(x_0; D) - E(x_0; D')| \leq 1$ and $|E| \geq L$, then $E$ satisfies $(\frac{1}{2}, \frac{1}{4\sigma^2 L^2} + \frac{1}{2L^2})$-Rényi Differential Privacy, where $\sigma^2$ depends on graph structure $\mathcal{G}$. Meanwhile, $F$ has no privacy guarantee, i.e., the Rényi differential privacy loss (8) is unbounded.*

*Proof.* Given the definition of function $E$ from above, we have that

$$\begin{aligned} E(x_0; D_{\mathcal{L}}) &= \sum_{v \in \mathcal{N}(x_0) \cap D_1} m(x_v; \theta(D_2)) + \sum_{w \in \mathcal{N}(x_0) \cap D_2} m(x_w; \theta(D_1)) \\ &= \sum_{v \in \mathcal{N}(x_0) \cap D_1} m(x_v \theta(D_2)) + \sum_{w \in \mathcal{N}(x_0) \cap D_2} m(x_w \theta(D_1)), \end{aligned} \tag{17}$$

where the second equality follows from the MLP assumption. Similarly for $F$ we have

$$\begin{aligned} F(x_0; D_{\mathcal{L}}) &= \sum_{v \in \mathcal{N}(x_0) \cap D_1} m(x_v; \theta(D_1)) + \sum_{w \in \mathcal{N}(x_0) \cap D_2} m(x_w; \theta(D_2)) \\ &= \sum_{v \in \mathcal{N}(x_0) \cap D_1} m(x_v \theta(D_1)) + \sum_{w \in \mathcal{N}(x_0) \cap D_2} m(x_w \theta(D_2)). \end{aligned} \tag{18}$$

We now define the adjacent datasets $D$ and $D'$ as follows. Assume $D = D_1$; one data point $\{x', y'\}$ is then randomly selected from Chunk I and removed $\{x', y'\}$ from $D_1$ forming $D' = D_1 \backslash \{x', y'\}$. Meanwhile, the

**Table 6:** Training time tested on AWS g4dn.12xlarge machine.

| Dataset | Base Model | Time(s) |
|---|---|---|
| House | GBM, NN | 52 |
| County | GBM, NN | 18 |
| VK | GBM, NN | 119 |
| Avazu | GBM, NN | 15 |
| OGB-Arxiv | NN | 199 |
| OGB-Products | NN | 837 |

unlabeled set $D_{\mathcal{U}}$ and $D_2$ remain the same. Our goal is to examine the extent to which $\boldsymbol{E}$ and $\boldsymbol{F}$ may leak information pertaining to $\{\boldsymbol{x}', \boldsymbol{y}'\}$ when $\{\boldsymbol{x}', \boldsymbol{y}'\}$ is removed from $D_1$ as described above.

Denote $\boldsymbol{x}_v, \boldsymbol{x}_w$ as training data in chunk I and chunk II. Assume $\begin{pmatrix} \boldsymbol{x}_v \\ \boldsymbol{x}_w \end{pmatrix}$ is drawn from a multivariate Gaussian distribution:

$$\begin{pmatrix} \boldsymbol{x}_v \\ \boldsymbol{x}_w \end{pmatrix} \sim \mathcal{N}\left( \begin{bmatrix} \mathbf{0} \\ \mathbf{0} \end{bmatrix}, \quad \begin{bmatrix} \boldsymbol{\Gamma}_{vv} & \boldsymbol{\Gamma}_{vw} \\ \boldsymbol{\Gamma}_{wv} & \boldsymbol{\Gamma}_{ww} \end{bmatrix}^{-1} \right), \tag{19}$$

where $\begin{bmatrix} \boldsymbol{\Gamma}_{vv} & \boldsymbol{\Gamma}_{vw} \\ \boldsymbol{\Gamma}_{wv} & \boldsymbol{\Gamma}_{ww} \end{bmatrix} = a\boldsymbol{I} + b\boldsymbol{N}$, $\boldsymbol{I}$ is identical matrix, $\boldsymbol{N}$ is normalized graph Laplacian, $a$ controls noise level and $b$ controls the smoothness over the whole graph.

From Proposition 5, the condition distribution of $\boldsymbol{x}_w$ given $\boldsymbol{x}_v = \boldsymbol{x}_v$ can be written as

$$\boldsymbol{x}_w | \boldsymbol{x}_v = \boldsymbol{x}_v \sim \mathcal{N}(-\boldsymbol{\Gamma}_{ww}^{-1}\boldsymbol{\Gamma}_{wv}\boldsymbol{x}_v, \boldsymbol{\Gamma}_{ww}^{-1}).$$

Condition on the data $D_1$, the distribution of $D_2$ is a conditional multivariate Gaussian distribution with mean $-\boldsymbol{\Gamma}_{ww}^{-1}\boldsymbol{\Gamma}_{wv}\boldsymbol{x}_v$ and variance $\boldsymbol{\Gamma}_{ww}^{-1}$. Furthermore, multiplicative Gaussian distribution $\boldsymbol{x}_w\boldsymbol{\theta}(D_1)$ introduces a Gaussian random noise into (17). According to Proposition (1) and (3), $\boldsymbol{E}$ satisfies $(\frac{1}{2}, \frac{1}{4\sigma^2 L^2} + \frac{1}{2L^2})$-Rényi Differential Privacy, where $\sigma^2$ depends on $\Gamma_{ww}^{-1}$ decided by graph structure.

Meanwhile, although (18) is deterministic, we can manually add Gaussian noise $\mathcal{N}(0, \sigma^2)$ such that $\boldsymbol{F}$ satisfies $\frac{\alpha}{2\sigma^2}$-Rényi Differential Privacy via Proposition (4). However, if we then let $\sigma \to 0$ to reproduce $\boldsymbol{F}$, we have that $\frac{\alpha}{2\sigma^2} \to \infty$, indicating that in fact $\boldsymbol{F}$ has no privacy guarantee. $\qquad\square$

## B  Experiment Details

### B.1  Data descriptions

**House**: node features are the property of house, edges connect the neighbors, the task is to predict the price of the house. **County**: each node is a county and edges connect two counties sharing a border, the task is to predict the unemployment rate for a county. **VK**: each node is a person and edges connect two people based on the friendships, the task is to predict the age of each person. **Avazu**: each node is a device and edges connect two devices if they appear on the same site with the same application, the target is the click-through-rate of a node. For **House**, **County**, **VK** and **Avazu** datasets, Training/validation/testing are randomly split with 6/2/2 ratio and all experiments results are averaged over 5 trails.

OGB-Arxiv, OGB-Products are standard datasets from OGB-leaderboards and all training/validation/testing splits follow the standard data splitting from OGB-leaderboards. Reddit is standard datset from Deep Graph Library (DGL).

**Table 7:** Dataset Statistics (- in Number of Classes means regression task)

| DATASET | NUMBER OF NODES | NUMBER OF EDGES | NUMBER OF CLASSES |
|---------|-----------------|-----------------|-------------------|
| HOUSE | 20,640 | 182,146 | - |
| COUNTY | 3,217 | 12,684 | - |
| VK | 54,028 | 213,644 | - |
| AVAZU | 1,297 | 54,364 | - |
| OGB-ARXIV | 169,343 | 1,166,243 | 40 |
| OGB-PRODUCTS | 2,449,029 | 61,859,140 | 47 |

**Table 8:** Hyperparameters

| DATASET | $\lambda$ | INPUT FOR STACKING LAYER |
|---------|-----------|--------------------------|
| HOUSE/COUNTY/VK/AVAZU | 0.9 | $(\boldsymbol{X}, \{\boldsymbol{F}_m^{(0)}, \boldsymbol{F}_m^{(1)}, \boldsymbol{F}_m^{(2)}, \boldsymbol{F}_m^{(3)}, \boldsymbol{F}_m^{(4)}, \boldsymbol{F}_m^{(5)}\})$ |
| OGB-ARXIV | 0.95 | $(\boldsymbol{X}, \{\boldsymbol{F}_m^{(0)}, \boldsymbol{F}_m^{(1)}, \boldsymbol{F}_m^{(3)}, \boldsymbol{F}_m^{(5)}, \boldsymbol{F}_m^{(7)}, \boldsymbol{F}_m^{(9)}\})$ |
| OGB-PRODUCTS | 0.97 | $(\boldsymbol{X}, \{\boldsymbol{F}_m^{(0)}, \boldsymbol{F}_m^{(1)}, \boldsymbol{F}_m^{(3)}, \boldsymbol{F}_m^{(5)}, \boldsymbol{F}_m^{(7)}, \boldsymbol{F}_m^{(9)}\})$ |

**Table 9:** Hyperparameters for C&S

| DATASET | $\lambda_1$ | KERNEL TYPE | $\lambda_2$ | KERNEL TYPE | NUM_PROPAGATION |
|---------|-------------|-------------|-------------|-------------|------------------|
| HOUSE/COUNTY/AVAZU | 0.8 | DA | 0.5 | DA | 5 |
| VK | 0.8 | DA | - | - | 5 |
| OGB-ARXIV | 0.9 | DA | 0.1 | AD | 50 |
| OGB-PRODUCTS | 0.3 | DAD | 0.3 | AD | 50 |

| $k$ | 2 | 3 | 4 |
|-----|---|---|---|
| Test Acc. | $0.710 \pm 0.001$ | $0.708 \pm 0.002$ | $0.712 \pm 0.001$ |

**Table 10:** Ablation study for $k$.

## B.2 Parameters for Graph-aware propagation

We do graph-aware propagation for the prediction to incorporate the graph structure. Table 8 shows two hyperparameters considered in the propagation part: weight $\lambda$ and number of propagation step $T$. We also present the hyperparameters for Correct and Smooth in Table 9.

## C Additional Ablations

The following tables show the performance with respect to $k$ and $L$ (refer to Table 10 and 11) . (Experiments on OGB-Arxiv with numerical embedding). From the following table, when $L \geq 2$, the outcome is not significantly affected by the value of $L$ (owing to the incorporation of the graph signal from the second layer). Therefore, we typically opt for $L = 2$. Typically, the Repeat bagging ($N$) is set to 1, which is the default value in the AutoGluon package. While a higher $N$ might enhance performance, it also significantly raises computational cost. Notice here, for both $L$ and $N$, in the main paper, we present a more generalized framework for our approach, which explains why we employ a for loop. The base models are determined by the input data type (e.g, Bert for text node feature and Tree-base model for tabular data). In our study, the

| $L$ | 1 | 2 | 3 | 4 |
|---|---|---|---|---|
| Test Acc. | $0.550 \pm 0.01$ | $0.712 \pm 0.02$ | $0.711 \pm 0.03$ | $0.707 \pm 0.01$ |

**Table 11:** Ablation study for $L$.

| $\lambda$ | 1.0 | 0.9 | 0.8 | 0.7 |
|---|---|---|---|---|
| Test Acc. | $0.707 \pm 0.01$ | $0.706 \pm 0.02$ | $0.697 \pm 0.02$ | $0.694 \pm 0.02$ |

**Table 12:** Ablation study for $\lambda$.

base models were designed in such a way that they are specific to a single modality, we do not do ablation study on them. Table 12 shows the performance is not sensitive to $\lambda$ (Experiments on OGB-Arxiv with numerical embedding).

