# OpenReview forum: "Graph Neural Networks Formed via Layer-wise Ensembles of Heterogeneous Base Models"
_TMLR — Accepted by TMLR_

### Review · Reviewer_gZTB · 2023-08-04

**Summary Of Contributions:**

To deal with heterogeneous node features (e.g., table and text) over graph-structured data, the main idea of this paper is to use bagging and stacking strategies with multiple models that are designed for such heterogeneous data. Also, to capture the graph structure (i.e., connections between nodes), the authors propose graph-aware feature propagation, which allows the model to consider the features of neighboring nodes of each node based on the energy function for graph-structured data. The authors evaluate the proposed BestowGNN on multiple benchmark datasets for node classifications and regressions with tabular, text, and numerical node features, demonstrating that the proposed BestowGNN outperforms other relevant ensemble baselines that usually cannot be aware of the graph structures.

**Audience:**

Yes

**Broader Impact Concerns:**

While the authors do not clearly discuss the broader impact, I don't have any concerns on it.

**Claims And Evidence:**

Yes

**Requested Changes:**

For major requested changes, please see the points in Weaknesses above.

Additionally,
* When discussing the label leakage in Section 2.1, it may be beneficial to more clearly describe what is this issue and why it happens.
* I am wondering why the authors do not use the same baselines across different datasets. For example, the authors compare the powerful baseline, namely AutoGluon + C&S, in Table 1; meanwhile, this baseline is not compared in Table 2.
* In Table 5, when reporting the time efficiency of the proposed model, it is more beneficial to include the other baseline models (e.g., the model that does not use the ensemble technique), for understanding relative efficiencies in comparisons to others.
* The paragraph (Base models) in Section 6 may appear earlier in the section. In particular, this paragraph explains the experimental setup of the proposed BestowGNN (e.g., which base models that they use for the ensemble), which may appear before showing the results.

**Strengths And Weaknesses:**

### Strengths
* The ensemble methods, including stacking and bagging, are rarely studied with graph-structured data.
* The idea of capturing the graph structures of raw text and tabular node features with graph-aware propagation is reasonable.
* The authors provide a meaningful theoretical analysis that the bagging strategy in the proposed BestowGNN can alleviate the label leakage (i.e., over-fitting) issue by connecting it to the Renyi Differential Privacy.
* The proposed BestowGNN outperforms relevant baselines on challenging datasets, such as OGB-arXiv and OGB-Products.

### Weaknesses
* The novelty of this paper is moderate. In terms of the stacking and bagging techniques, the authors directly use the existing methods. While the authors propose to consider the graph structures with graph-aware propagation during ensemble, this is a general way other studies also use when they deal with graph-structured data.
* In Table 1, when considering the standard deviations, the performance improvement of the proposed BestowGNN is marginal against the baseline (AutoGluon + C&S), and the possible reason for this is not clearly discussed. In addition to this, it would be meaningful to discuss the difference between the AutoGluon + C&S and the proposed BestowGNN in terms of the methodology, since AutoGluon + C&S is the ensemble baseline that can also be aware of the graph structures similar to the proposed method.
* In Table 3, the performance of the BestowGNN is far from the state-of-the-art results in the OGB leaderboards; therefore, it would be meaningful to discuss why the gaps between the proposed method and the state-of-the-art methods are significant and how we can bridge those gaps.

---

> ### Author Response · Authors · 2023-10-26
> **Response (Part 1)**
>
> Thanks for the constructive comments.  We have responded below to each point, as well as updated the draft to reflect these comments.  Areas of significant change have also been highlighted in red.
>
> **Question:** *The novelty of this paper is moderate.*
>
> **Answer:** We agree with the reviewer that the constituent components of our framework have been previously studied in the literature, and so there is little novelty when we view them in isolation.  That being said, we would argue that there are other key dimensions of novelty along which our system provides notable contributions.  We summarize these dimensions as follows:
> 1. Even if the individual components are not novel, selecting appropriate components (from the near limitless space of possibilities) and forming a cohesive system to successfully address unsolved modeling challenges is nonetheless a valuable form of innovation.  In fact many popular ML frameworks rely heavily on prior work in this way.  With respect to our submission, we uniquely integrate bagging, stacking, and label propagation leading to the first AutoML system that can cope with arbitrary graph datasets with heterogeneous feature-types (numerical, categorical, text).
>
> 2. There is additional novelty in the empirical demonstration that a single unified architecture, with various modular/interchangeable base learners, is capable of consistently achieving competitive performance across disparate graph datasets relative to existing alternatives with fundamentally different dataset-specific architectures.
>
> 3. Finally, novelty can acrue through supporting analysis or new ways of viewing a given system, whether or not the given system itself is new.  In our case, we formalize how the proposed bagging and stacking framework can effectively mitigate label leakage issues within the graph setting through an innovative use of differential privacy tools.
>
>
> **Question:** *In Table 1 ... the performance improvement of the proposed BestowGNN is marginal against the baseline (AutoGluon + C&S), and the possible reason for this is not clearly discussed. In addition to this, it would be meaningful to discuss the difference between the AutoGluon + C&S and the proposed BestowGNN in terms of the methodology, since AutoGluon + C&S is the ensemble baseline that can also be aware of the graph structures similar to the proposed method.*
>
> **Answer:** AutoGluon+C&S performs Correct-and-Smooth as a post-processing step on top of Autogluon's predictions, in order to provide a post hoc account for the graph structure during inference. In contrast, our method BestowGNN applies graph-aware propagation *within* the middle of stacking layers to incorporate the graph information. This means that from the second layer on, all base learners can see the graph information through this training-time graph propagation.
>
> That being said, we agree that in Table 1, BestowGNN is only marginally better against the baseline AutoGluon+C&S on the individual datasets.  However, the improvement becomes more significant when aggregating across all datasets in Table 1, and further so when aggregating across all benchmarks in our paper (noting that in some cases AutoGluon+C&S is not competitive at all, e.g., OGB-Products). Returning to Table 1, the reason the gap between BestowGNN and AutoGluon+C&S is not larger is likely because these are relatively small datasets whereby simple models can be adequate (and more challenging tabular/graph benchmarks are not readily available, despite their ubiquity in industry).

---

> > ### Author Response · Authors · 2023-10-26
> > **Response (Part 2)**
> >
> > **Question:** *In Table 3, the performance of the BestowGNN is far from the state-of-the-art results in the OGB leaderboards; therefore, it would be meaningful to discuss why the gaps between the proposed method and the state-of-the-art methods are significant and how we can bridge those gaps.*
> >
> > **Answer**: When we originally started our experiments (which was quite some time ago), BestowGNN performed competitively against top baselines on the OGB leaderboards. Moreover, our original aim was not to directly surpass all the SOTA results, which vary over time. Instead our objective was to achieve competitive results using a single, unified architecture; only the base model for processing input features for each dataset may be different.  In this regard, we were motivated by the fact that quite different GNN models, or compositions of multiple GNN models and training heuristics, often dominate competitive leaderboards for node classification/regression tasks, e.g., on OGB and elsewhere completely different network architectures (not just different base models per se) occupy the top positions for different datasets and data types.  Notably, the top model for one is unlikely to transfer elsewhere with similar success.
> >
> > Nonetheless, we realize that the leaderboard has now changed significantly, and we have updated our results, particularly those in Table 3 of the revision.  Specifically, for results based on raw text features (which are better than the original compressed OGB features), we only include the rank-1 model from OGB-Arxiv and the rank-1 model from OGB-Products (at the time of our submission).  However, we apply both of these models to both of the datasets to illustrate the reduction in performance upon transfer with standard hyperparameter tuning (despite that fact that both of these models are designed to handle text features).  In brief here, the top OGB-Arxiv model, TAPE+RevGAT, achieves only 82.16 accuracy on OGB-Products, placing it 30th on the Products leaderboard.  Similarly, the top Products model, a composite architecture called GIANT-XRT+GAMLP+MCR, achieves 75.5 accuracy on OGB-Arxiv, which drops to 16th place.  Consequently, even for datasets with similar node features, a situation more amenable to model transfer, the performance drop can be significant.  And yet in both of these cases, our generic BestowGNN architecture performs competitively.
> >
> >
> > We also note that the top OGB models amount to various heuristic combinations of existing approaches, and if model selection were based on the validation set, for OGB-Products the 10th place model (GIANT-XRT+GAMLP+MCR) would be chosen (which has the highest validation accuracy).  And yet this model has similar validation and test accuracy to our generic BestowGNN approach.
> >
> > In any event, we hope this context is helpful, and we have revised our experimental section to reflect these considerations.
> >
> >
> >
> > **Question:** *When discussing the label leakage in Section 2.1, it may be beneficial to more clearly describe what is this issue and why it happens.*
> >
> > **Answer:** We have updated Section 2.1 to provide more details about label leakage when contextualizing the SAGN and GAMLP models, both of which use labels as model inputs during training and are at risk of label leakage.  Further discussion regarding label leakage, related to SAGN and GAMLP models that may experience it, and the impact of BestowGNN in resolving it are deferred to Section 5.
> >
> > **Question:** *Using the same baselines across different datasets. For example, the authors compare the powerful baseline, namely AutoGluon + C&S, in Table 1; meanwhile, this baseline is not compared in Table 2.*
> >
> > **Answer:** Thanks for the suggestion.  It was an oversight to omit AutoGluon+C&S, and it has now been added to a revision of Table 2.  That being said, AutoGluon+C&S performance is similar to AutoGluon, and worse than PCAPass+XGBoost, the top-performing baseline in Table 2.  We reproduce the revised table here for reference.
> >
> > Beyond this, the reason we do not use the same baselines uniformly across all datasets is because the top performing models vary considerably from benchmark to benchmark, and some are only designed for a particular input data type.  Indeed our purpose is to show that BestowGNN can be competitive in disperate regimes despite its conformity to a single core architecture.
> >
> > Updated version of Table 2:
> > | Method          | Acc.  |
> > |-----------------|-------|
> > | PCAPass+XGBoost | 96.26 |
> > | GraphSAGE       | 95.04 |
> > | AutoGluon       | 95.83 |
> > | AutoGluon+C&S   | 96.00 |
> > | BetowGNN        | 96.44 |

---

> > > ### Author Response · Authors · 2023-10-26
> > > **Response (Part 3)**
> > >
> > > **Question:** *When reporting the time efficiency of the proposed model, it is more beneficial to include the other baseline models.*
> > >
> > > **Answer:** Thank you for your suggestion. In the main paper, we focused on demonstrating the time efficiency of our approach, ensuring that our method's computational cost is within a reasonable range. We did not gather time efficiency data for other benchmarks because they possess varying code structures and package versions. This diversity makes it hard to uniformly gauge training/inference time or memory usage.
> > >
> > > **Question:** *The paragraph (Base models) in Section 6 may appear earlier in the section. In particular, this paragraph explains the experimental setup of the proposed BestowGNN (e.g., which base models that they use for the ensemble), which may appear before showing the results.*
> > >
> > > **Answer:** Thanks for the suggestion.  We have reorganized Section 6.

---

> > > > ### Comment · Reviewer_gZTB · 2023-11-06
> > > >
> > > > Thank you for your response. The authors clearly address my concerns on novelty, baselines, marginal performances, and other minor suggestions.

---

### Review · Reviewer_nFRC · 2023-08-06

**Summary Of Contributions:**

This paper introduces BestowGNN, a generic unified method of combining standard, off-the-shelf, point-wise classifiers with graph propagation, to enhance the predictions on graph data. The idea of enabling the combination of such ideas is clearly motivated by prior successes of methodologies that compose pointwise classifiers with graph smoothing operators, such as BGNN and C&S. The framework that the authors propose is as follows:

* Repeatedly resample the training dataset into held-out splits using bagging.
* Train all relevant base models on each split within each bagging run.
* Combine the predictions of each base model on their held-out nodes, as well as the overall validation / testing nodes for the task.
* Perform a few steps of graph propagation (using energy minimisation) to smooth out these predictions.
* Re-run this procedure K times, at each step using the concatenated outputs of graph propagation at various scales as input features.

Theoretical motivation for these design choices is provided, and favourable performance is demonstrated on relevant datasets.

**Audience:**

Yes

**Broader Impact Concerns:**

No strong concerns to report; however, I did feel like the paper could have taken more rigour in describing the computational footprint of BestowGNN, especially considering the nested-ness of its forward pass. In the very least, I would have hoped to see a rigorous time complexity analysis, and supplementing Table 5 with representative baseline model (e.g. off-the-shelf GNN) timings, to empirically compare any slowdowns or speedups induced.

**Claims And Evidence:**

No

**Requested Changes:**

1. BestowGNN is certainly not the first method to propose combining pointwise classifiers with graph operators. This is implied by the authors as well, by making references to works like BGNN and C&S. However, all of these references are made in passing, and certain variants of these models are then compared to in the experiments. The paper would strongly benefit from a detailed subsection elaborating the similarities and differences that BestowGNN has to both of these models, and ideally a discussion of possible tradeoffs between them.

2. The paper seems to defend its choice of energy minimisation-based propagation by invoking the 'oversmoothing' problem. But, at least from my scan of the paper, I saw no clear indication (or proof) that the authors' approach makes oversmoothing less of an issue. This claim should be either toned down, or more explicitly and rigorously backed up by theoretical evidence. Further, in the very least, I would expect to see the following two empirical comparisons to solidify this claim:
    * To confirm the lack of suitability of message-passing style propagation, it would be useful to provide a comparison against a variant of BestowGNN which uses a model like SGC (a nonparametric GNN) to propagate information.
    * I have doubts that the tasks the authors considered would be appropriate to show clear impact of the oversmoothing effect in the first place, as at least some of these datasets are known to be homophilic. It would be highly beneficial to evaluate BestowGNN on heterophilic benchmarks---perhaps from the recent work of Platonov et al. (ICLR'23): https://openreview.net/forum?id=tJbbQfw-5wv.

3. From the experiments presented, especially on the OGB benchmarks, I cannot draw clear conclusions about how does BestowGNN compare against the state-of-the-art. A simple look at the OGB leaderboards (https://ogb.stanford.edu/docs/leader_nodeprop/) shows that there are variants of GIANT that would outperform all numbers in Table 3. Further, in both Table 1 and 3, the authors often bold-face only the BestowGNN result, even though there are several other baselines that are clearly with highly overlapping error bars to BestowGNN, casting doubt at the statistical significance of its outperformance. I am not saying that the paper needs to set state-of-the-art to be publishable in TMLR---the argument the authors make about the uniformness of the method is certainly on point!---but the paper also needs to make a clear message about when the method is clearly outperforming its baselines, and when there exist better-performing methods. The correctness and clarity of this information would be invaluable to a GNN practitioner hoping to potentially make advantage of this method in their work.

4. Last, but certainly not least, BestowGNN is a complex system consisting of several mechanisms stacked on top (each, roughly, exemplified by a for loop in Algorithm 1). Naturally, it is important to empirically defend the inclusion of these systems through careful ablation studies. Table 4 does this for the number of propagation steps, and the inclusion of bagging---both are highly appreciated. But there are several other axes as well: stacking ($L$), repeated bagging ($N$) and multiple models ($|\mathcal{M}|$). It would be very useful if similar tables to Table 4 could be provided, showing the impact of various amounts of repeated bagging, various numbers of stacked layers, and the specific combinations of models used. As currently presented, unless I am missing something, I do not see any convincing evidence that either of these for-loops are necessary---and it would be great to have that evidence: both if you're a practitioner and if you are a researcher extending the method.

I hope my suggestions are useful to the authors. I am willing to champion the paper if they are appropriately addressed.

**Strengths And Weaknesses:**

The paper's proposal is clearly explained, well-motivated, and timely. The experiments are, in principle, well-designed, and the model appears to perform competitively against relevant baselines.

However, while the paper certainly has merit, and could be a useful TMLR publication, there are also several issues in the work's presentation and the depth of its experimentation, which could be misleading to uninitiated readers. Therefore, several changes would need to be implemented before I would be comfortable recommending acceptance. Such issues include:

* Lack of appropriate positioning of BestowGNN with respect to prior art. This is all the more important, given that the method relies on a complex combination of existing ideas.
* Unclear and somewhat 'cavalier' handling of the oversmoothing problem in the paper.
* I have some doubts about possible cherry-picking of the baselines, as well as the statistical significance of certain claims the paper makes.
* Lack of depth in the ablation studies.

Please see the "Requested Changes" cell for a summary of my suggestions to the authors.

---

> ### Author Response · Authors · 2023-10-26
> **Response (Part 1)**
>
> Thanks for the constructive comments.  We have responded below to each point, as well as updated the draft to reflect these comments.  Areas of significant change have also been highlighted in red.
>
>
> **Question:** *Lack of appropriate positioning of BestowGNN with respect to prior art. This is all the more important, given that the method relies on a complex combination of existing ideas.*
>
> **Answer:** We have added  more discussion about the similarities and differences between BestowGNN and other prior work in the polished version of the related work section (marked as red).
>
> **Question:** Unclear and somewhat 'cavalier' handling of the oversmoothing problem in the paper.
>
> **Answer:** Just to clarify, that graph propagation operators based on energy minimization can alleviate oversmoothing is not an original claim we intend to make ourselves.  Rather, this claim spreads across a wide variety of prior work (e.g., see the references at the beginning of Section 3.2), and has already been supported by both theory and experiment.  In particular, once we establish that minimizers of an energy function do not oversmooth, and graph propagation steps merely push closer to such a minimizer, then oversmoothing cannot actually pose a risk.  This can also be empirically verified, via experiments similar to what the reviewer suggests.
>
> As a representative example, Figure 2 in reference [1] shows graph propagation steps vs accuracy of an energy minimization method vs SGC exactly as the reviewer mentioned.  From this figure we observe that the former is quite stable while with the latter performance plummets quickly.  Similar, this same reference shows strong performance on heterophily datasets in Table 3 as the reviewer suggested trying as well. As such, we have added further citations to this effect within our paper to better support these claims (highlighted in red in Sections 3.2 and 4.5), and apologize that our original submission was not sufficiently clear.
>
> [1]. Graph neural networks inspired by classical iterative algorithms
>
>
>
>
> **Question:** *Have some doubts about possible cherry-picking of the baselines ... cannot draw clear conclusions about how BestowGNN compares against SOTA.*
>
>
> **Answer**: When we originally started our experiments (which was quite some time ago), BestowGNN performed competitively against top baselines on the OGB leaderboards. Moreover, our original aim was not to directly surpass all the SOTA results, which vary over time. Instead our objective was to achieve competitive results using a single, unified architecture; only the base model for processing input features for each dataset may be different.  In this regard, we were motivated by the fact that quite different GNN models, or compositions of multiple GNN models and training heuristics, often dominate competitive leaderboards for node classification/regression tasks, e.g., on OGB and elsewhere completely different network architectures (not just different base models per se) occupy the top positions for different datasets and data types.  Notably, the top model for one is unlikely to transfer elsewhere with similar success.
>
> Nonetheless, we realize that the leaderboard has now changed significantly, and we have updated our results, particularly those in Table 3 of the revision.  Specifically, for results based on raw text features (which are better than the original compressed OGB features), we only include the rank-1 model from OGB-Arxiv and the rank-1 model from OGB-Products (at the time of our submission).  However, we apply both of these models to both of the datasets to illustrate the reduction in performance upon transfer with standard hyperparameter tuning (despite that fact that both of these models are designed to handle text features).  In brief here, the top OGB-Arxiv model, TAPE+RevGAT, achieves only 82.16 accuracy on OGB-Products, placing it 30th on the Products leaderboard.  Similarly, the top Products model, a composite architecture called GIANT-XRT+GAMLP+MCR, achieves 75.5 accuracy on OGB-Arxiv, which drops to 16th place.  Consequently, even for datasets with similar node features, a situation more amenable to model transfer, the performance drop can be significant.  And yet in both of these cases, our generic BestowGNN architecture performs competitively.
>
>
> We also note that the top OGB models amount to various heuristic combinations of existing approaches, and if model selection were based on the validation set, for OGB-Products the 10th place model (GIANT-XRT+GAMLP+MCR) would be chosen (which has the highest validation accuracy).  And yet this model has similar validation and test accuracy to our generic BestowGNN approach.
>
> In any event, we hope this context is helpful, and we have revised our experimental section to reflect these considerations.

---

> > ### Author Response · Authors · 2023-10-26
> > **Response (Part 2)**
> >
> > **Question:** *have some doubts about ... the statistical significance of certain claims the paper makes.*
> >
> >
> > It is true that on any given benchmark there may be other baselines with performance similar to BestowGNN, e.g.,  retrofitting a top performing tabular base model AutoGluon with C&S graph propagation is competitive on the tabular baselines in Table 1.  But the larger point is that no single model performs well across all tasks, e.g., AutoGluon+C&S performs poorly on OGB-Products in Table 3.  So the statistical significance becomes more pronounced when considering results in aggregate across all tasks and benchmarks, where BestowGNN is consistently at the top.
> >
> >
> > **Question:** *Lack of depth in the ablation studies.*
> >
> >
> >
> > **Answer**:  Additional ablations per the reviewer's request have been added to the appendix (see red-colored content there).  We summarize some of these results here: The table below shows the performance with respect to $L$. (Experiments on OGB-Arxiv with numerical embedding). From the following table, when $L \geq 2$, the outcome is not significantly affected by the value of $L$ (owing to the incorporation of the graph signal from the second layer). Therefore, we typically opt for $L=2$.
> >
> > | $L$       |1      | 2     | 3     | 4     |
> > |-----------|-------|-------|-------|-------|
> > | Test Acc. | 0.550     | 0.712 | 0.711 | 0.707 |
> >
> > Additionally, the Repeat bagging $(N)$ is typically set to 1, which is the default value in the AutoGluon package. While a higher $N$ might enhance performance, it also significantly raises computational cost. Notice here, for both $L$ and $N$, in the main paper, we present a more generalized framework for our approach, which explains why we employ a for loop.
> >
> > The base models are determined by the input data type (e.g, Bert for text node feature and Tree-base model for tabulr data). Thus the selsection of base models cannot be regraded as hyperparameters.  Beyond this, please see the appendix for more details.

---

> ### Comment · Reviewer_nFRC · 2023-12-23
> **Thank you. Some issues still remain.**
>
> Thank you for your response which addresses most of my concerns. Some issues still remain, however.
>
> Re- **Q1**, I need to draw to your attention the following new passage:
>
> _"While C&S can in principle be combined with any base learner aligned with available node features, the base learner itself does not have access to graph structure during training. This is unlike BestowGNN, whereby graph propagated signals are passed through trainable layers as ensembled models are stacked together."_
>
> which, as currently written, is wrong. There is nothing about C&S which requires the base learner to not use the graph structure. In fact, the C&S paper itself shows state-of-the-art results of models like "GAT + C&S", where the base model is a graph attention network. Please correct this sentence.
>
> Re- **Q2**, I still think that having explicit experiments combining SGC with your method would be valuable for the readers. But, in light of you pointing out works like Yang et al., I will not hold this as a bar for acceptance into TMLR.
>
> Re- **Q3**, I do agree with the Authors' claim that, even if we cannot show statistically significant outperformance on each dataset, showing competitive performance on many of them can in and of itself be significant. However, reading the revised paper, I see my comment was addressed by _bold-facing top-2 methods_, which is not what I asked for. I asked to _bold-face **all** methods that have indistinguishable performance to the best one (e.g. when the best mean is within one-sigma of the given mean)_. This should make it more clear which results could be seen as significant. At present, this is a very unconventional use of bold-face, and it is likely to confuse.
>
> Re- **Q4**, thank you for adding more hyperparameter sweeps. I have two follow-ups:
> * It would have been very useful to provide error bars on these results, to get a feel about the stability of the results as you modify the hyperparameter.
> * I disagree with the phrasing _"The base models are determined by the input data type... thus the selection of base models cannot be regraded as hyperparameters."_. Are the models the Authors study the only possible deep learning models? Can models like Transformers not be applied to any data (including tabular) if set up correctly?
>
> I am okay with not including ablations on base model choices, but the justification needs to be written in a more proper way, for example, that the way you set up the base models used in your study, they can only ever be applied to one of the modalities, and hence you cannot sweep them.

---

> > ### Author Response · Authors · 2024-01-03
> > **Response**
> >
> > Thank you for the additional feedback. The holiday period has caused some delay in our response; however, we are now available to quickly address these follow-up points as follows.
> >
> > Regarding Q1, Q2, and Q3, these are straightforward to incorporate into a final revision of our paper.
> >
> > As for Q4, we actually have computed the error bars the reviewer requested, and they are all quite small (on the order of 0.001 to 0.01) such that all of the ablation results and conclusions are quite stable.  We can easily add these to the corresponding tables in the Appendix.  And finally, we can also clarify the descriptions/selections of our base models in the revision, noting that it is in fact quite easy to swap in alternative base models into our modular framework, including Transformers for tabular data if desired.  That being said, tree-based models were more effective on tabular data when we conducted our experiments, although the situation could of course change as newer base models and training procedures are developed.

---

### Review · Reviewer_QGQx · 2023-12-06

**Summary Of Contributions:**

Summary Of Contributions
The paper designs an innovative framework , i.e. BestowGNN, that is capable of leverages bagging and stacking strategies to GNNs. It theoretically reduces label leakage problem and shows strong performance on multiple types of graphs.

**Audience:**

Yes

**Broader Impact Concerns:**

Broader Impact Concerns
None

**Claims And Evidence:**

Yes

**Requested Changes:**

Requested Changes
See W2

**Strengths And Weaknesses:**

Strengths And Weaknesses
Strengths:

Innovative and Universal System Design: The paper introduces a framework capable of leveraging diverse collections of models for heterogeneous graph datasets with various feature types such as numerical, categorical, and text. This versatility is a significant advancement in the field.

Theoretical Insights: The work makes a substantial theoretical contribution by exploring the label leakage issue in label-augmented Graph Neural Networks (GNNs), offering new perspectives and understanding.

Robust Experimental Performance: The paper demonstrates strong performance through a series of comprehensive experiments, showcasing the practical effectiveness of the proposed system on graphs with numerical, categorical, and text features.

Weaknesses:

Increased Complexity: The application of both bagging and stacking techniques introduces additional complexity, which could potentially hinder the practical deployment and usability of the system.

Manuscript Quality: The current state of the manuscript necessitates improvements. Notably, there are missing references, as indicated by "?" marks on page 9. Additionally, the heavy use of notation throughout the paper detracts from its overall readability, suggesting a need for clearer exposition and possibly simplification.

---

> ### Author Response · Authors · 2023-12-20
> **Response**
>
> Thanks for the constructive comments.  We have responded below to each point, as well as updated the draft to reflect these comments.
>
> **Question:** Increased Complexity: The application of both bagging and stacking techniques introduces additional complexity, which could potentially hinder the practical deployment and usability of the system.
>
> **Answer:** As a stacking model, our complexity is analogous to layer-wise training, i.e., no need to repeatedly compute the expensive forward and backward propagation over the graph at each epoch as when training a regular GNN. Additionally, we have addressed the time efficiency of our approach in the main paper, demonstrating that the computational cost is within a reasonable range. We also remark that by basing relevant portions of our implementation on AutoGluon, a highly developed open-source package for training machine learning models, we enhance the practicality and usability of our proposed approach.
>
>
>
>
> **Question:** Manuscript Quality: The current state of the manuscript necessitates improvements. Notably, there are missing references, as indicated by "?" marks on page 9. Additionally, the heavy use of notation throughout the paper detracts from its overall readability, suggesting a need for clearer exposition and possibly simplification.
>
> **Answer:** Thanks for the suggestion, and we have corrected the typos the reviewer pointed out. Regarding the notation, we find it challenging to precisely convey the technical aspects of the proposed method without it; however, we can try to make as readable as possible.

---

### Review · Reviewer_QmtH · 2023-12-12

**Summary Of Contributions:**

This paper proposes BestowGNN, a novel framework that integrates graph-aware propagation with ensembles of arbitrary models intended for independent and identically distributed (IID) data. This approach addresses limitations of Graph Neural Networks (GNNs) for dealing with non-numerical node features like text or tabular data. By leveraging bagging and stacking strategies, BestowGNN effectively mitigates label leakage and overfitting, achieving strong generalization across various graph datasets.

**Audience:**

Yes

**Claims And Evidence:**

Yes

**Requested Changes:**

See the weakness.

**Strengths And Weaknesses:**

Strengths:
* Innovative Integration: Combines graph-aware propagation with models designed for IID data, addressing GNNs' limitations with non-numerical data.
* Mitigation of Label Leakage: Introduces techniques to effectively handle label leakage, a common issue in layer-wise training strategies.
* Generalization Across Datasets: Demonstrates strong performance across a variety of graph datasets, showcasing the method's adaptability.
* Theoretical Analysis: Provides a thorough theoretical analysis of the proposed methods, enhancing the paper's credibility.
* Empirical Validation: Empirical results support the framework's effectiveness, with comparisons to state-of-the-art methods.

Weaknesses:
* Lack of Statistics of Datasets: The paper does not present the detailed statistics of the datasets used in the paper. Providing these dataset statistics, such as the number of nodes, edges, features, and classes in graph datasets, is crucial for readers to fully understand the context and scale at which the proposed methods are tested. This information is essential for replicating the study, evaluating the model's performance in various scenarios, and comparing it with other methods. Without these statistics, it's challenging to asses the robustness and adaptability of the proposed framework to different types of graph data.

---

> ### Author Response · Authors · 2023-12-20
> **Response**
>
> Thanks for the constructive comments.  We have responded below to each point, as well as updated the draft to reflect these comments.
>
> **Question:** Lack of Statistics of Datasets
>
> **Answer:** Thanks for the suggestion; we have added the data statistics table in the revised version.

---

### Decision · Action_Editor_sM7S · 2024-01-22

**Recommendation:** Accept with minor revision

**Comment:**

The paper aims to expand joint training of Graph Neural Networks (GNN) with complex feature extraction models, which maynot be neural networks, such as decision trees, and not amenable to training by direct backpropogation. The paper presents BestowGNN, a framework that integrates graph-aware propagation with ensembles of models designed for independent and identically distributed (IID) data. In particular, the paper considers node features like numerical, text, and tabular data. In the initial draft the presentation and writing of the paper leaves more to be desired. We thank the authors and reviewers to actively engage in discussion for making the paper better. During discussion many of the reviewer concerns/questions were resolved like missing baselines and statistical significance of the result. There were some concerns left: 1) about oversmoothing claims, which can be made more rigorous more rigorous by more extensive experiments, 2) ablation for need of multiple heterogenous base models $\mathcal{M}$ vs using a transformer across modalities; 3) novelty, but it is not a criteria for TMLR.  Nevertheless, the proposed method and technique are correct, experimental result demonstrate competitive performance across various datasets using a single unified architecture with only the base model for processing input features varying for each dataset, and thus the approach will be of interest to the community, hence I propose to accept the paper with some minor modifications.

Please add all the reviewer suggestions in the final draft along with following two points:
- Revise the claim that C&S cannot use a graph-based base model or only in principle. The original C&S paper seems to do this.
- Consistency of bold-facing the top performing methods. Please bold-face all methods which have a statistical overlap as it is not feasible to decide which one is better given current statistics.

**Audience:**

Yes, both graph learning and few-shot/zero-shot learning community will be interested in this paper.

**Claims And Evidence:**

Yes